# Associations of timing of physical activity with all-cause and cause-specific mortality in a prospective cohort study

Hongliang Feng [1,2,12], Lulu Yang [1,12], Yannis Yan Liang [1,12] ✉, Sizhi Ai[3,4,5], Yaping Liu[2,5], Yue Liu[1], Xinyi Jin[1], Binbin Lei[1], Jing Wang [3,5], Nana Zheng[3], Xinru Chen[1,3], Joey W. Y. Chan [2,5], Raymond Kim Wai Sum[6], Ngan Yin Chan[2,5], Xiao Tan[7,8], Christian Benedict [9], Yun Kwok Wing [2,5] & Jihui Zhang [3,5,10,11] ✉

There is a growing interest in the role of timing of daily behaviors in improving health. However, little is known about the optimal timing of physical activity to maximize health benefits. We perform a cohort study of 92,139 UK Biobank participants with valid accelerometer data and all-cause and cause-specific mortality outcomes, comprising over 7 years of median follow-up (638,825 person-years). Moderate-to-vigorous intensity physical activity (MVPA) at any time of day is associated with lower risks for all-cause, cardiovascular disease, and cancer mortality. In addition, compared with morning group (>50% of daily MVPA during 05:00-11:00), midday-afternoon (11:00-17:00) and mixed MVPA timing groups, but not evening group (17:00-24:00), have lower risks of all-cause and cardiovascular disease mortality. These protective associations are more pronounced among the elderly, males, less physically active participants, or those with preexisting cardiovascular diseases. Here, we show that MVPA timing may have the potential to improve public health.

National and international guidelines state that physical activity (PA) is one of the most important measures for individuals of all ages to improve health[1–3]. Compelling evidence suggests that PA is associated with lower risks of mortality and morbidities, including cardiovascular disease (CVD) and cancer[4,5]. Recent studies have further provided insights on maximizing the health benefits of PA through different dimensions, such as intensity and type (recreational or non-recreational)[5–7]. Although the health effects of timing of other daily behaviors (e.g., meals and sleep) are increasingly being recognized[8–11], little is known about the effect of timing of PA on health, which is a major public health interest[12].

[1]Guangdong Cardiovascular Institute, Guangdong Provincial People's Hospital, Guangdong Academy of Medical Sciences, Guangzhou, Guangdong, China. [2]Li Chiu Kong Family Sleep Assessment Unit, Department of Psychiatry, Faculty of Medicine, The Chinese University of Hong Kong, Shatin, Hong Kong SAR, China. [3]Center for Sleep and Circadian Medicine, The Affiliated Brain Hospital of Guangzhou Medical University, Guangzhou, Guangdong, China. [4]Department of Cardiology, Heart Center, The First Affiliated Hospital of Xinxiang Medical University, Weihui, Henan, China. [5]Department of Psychiatry, Faculty of Medicine, The Chinese University of Hong Kong, Hong Kong SAR, China. [6]Department of Sports Science and Physical Education, Faculty of Education, The Chinese University of Hong Kong, Shatin, Hong Kong SAR, China. [7]Department of Big Data in Health Science, Zhejiang University School of Public Health and Sir Run Run Shaw Hospital, Zhejiang University School of Medicine, Hangzhou, China. [8]Department of Medical Sciences, Uppsala University, Uppsala, Sweden. [9]Molecular Neuropharmacology, Department of Pharmaceutical Biosciences, Uppsala University, 751 24 Uppsala, Sweden. [10]Guangdong Mental Health Center, Guangdong Provincial People's Hospital (Guangdong Academy of Medical Sciences), Southern Medical University, Guangzhou, Guangdong, China. [11]Key Laboratory of Neurogenetics and Channelopathies of Guangdong Province and the Ministry of Education of China, Guangzhou Medical University, Guangzhou, China. [12]These authors contributed equally: Hongliang Feng, Lulu Yang, Yannis Yan Liang. ✉e-mail: liangyan@link.cuhk.edu.hk; zhangjihui@gzhmu.edu.cn

Animal studies have consistently found that the timing of PA may affect metabolic functions[13–15], whereas findings from human studies are inconclusive and contradictory[16–18]. For example, Sato et al. found a more robust metabolic effect of exercise in the morning than at night[14]. In line with this, a human experimental study found that morning exercise at 10:30 tends to be more effective than afternoon exercise at 16:30 in reducing hyperglycemia[18]. In contrast, other human experimental studies either found a greater glucose reduction in the afternoon or evening exercise groups[16,17] or no differences in glucose responses across morning, afternoon, and evening exercises[19]. Thus, the optimal time-of-day of PA to maximize the health benefits in humans, especially for long-term health, remains unclear. In addition, despite the extensive evidence supporting the role of PA at any intensity and type in reducing mortality risk[5–7], little is known about the association between PA at different times of day and health risk.

In this work, we focus on PA at a relatively high-intensity level (i.e., moderate-to-vigorous intensity PA, MVPA) to determine a robust PA timing phenotype[20] and aim to investigate the associations of MVPA timing with all-cause, CVD, and cancer mortality risk in the UK Biobank. The UK Biobank has collected seven-day activity data from over 100,000 participants by accelerometers, which can continuously record 24-h activity without interfering with an individual's daily routine[6,21]. Therefore, it provides an unprecedented opportunity to test the association between MVPA timing and health. We also explore whether some demographic and health-related characteristics, including age, sex, MVPA level, CVDs, and obesity, modify the associations between MVPA timing and mortality risk and investigate the health effect of MVPA at different times of the day. We find that midday-afternoon and mixed MVPA timing groups have lower risks for all-cause and CVD mortality, independent of MVPA level. These protective associations are more pronounced among the elderly, males, less active (i.e., below WHO PA recommendation) individuals, or those with preexisting CVDs. Interestingly, MVPA is associated with lower risks for all-cause, CVD, and cancer mortality, irrespective of the time of day. Our findings may help to improve strategy-making and practice in public health.

## Results

Over a median of 7.0 years (638,825 person-years), 3088 (3.35%) participants died from all-cause, 1076 (1.17%) from CVD, and 1872 (2.03%) from cancer. The baseline characteristics of 92,139 participants are shown in Table 1. Overall, four timing groups presented similar profiles for sociodemographic, lifestyle, and health status. However, the mixed group showed a higher Townsend deprivation index than other groups. The morning group had a lower education level than other groups. The evening and mixed groups tended to have shorter sleep duration, since they were more from the group of <7 hours/day than morning and midday-afternoon groups. The morning group were more from the group of sleep midpoint earlier than 02:30 than other groups. Additionally, we found that morning and evening groups had fewer MVPA minutes than the other two groups.

### Associations of total MVPA volume and MVPA within three time windows with mortality outcomes

Figure 1a–c show the associations between total MVPA volume and mortality outcomes adjusted for age, sex, ethnicity, Townsend deprivation index, recruitment center, education level, the season of accelerometer wear, smoking status, alcohol intake, healthy diet score, sleep duration, and sleep midpoint. MVPA (continuous) was associated with lower risks of all-cause, CVD, and cancer mortality (all $P_{overall} < 0.001$). These associations were nonlinear, being steeper between ~0–150 min/week, then stabilizing between ~150–750 min/week (all $P_{non-linear} < 0.001$). The beneficial associations between MVPA and mortality outcomes were consistent with previous reports[22–24].

Figure 1d–f show the associations between MVPA within three time windows and mortality outcomes. MVPA minutes within three time windows were all significantly associated with lower risks of all-cause, CVD, and cancer mortality (all $P_{overall} < 0.001$; all $P_{non-linear} < 0.001$), even after controlling for age, sex, ethnicity, Townsend deprivation index, recruitment center, education level, the season of accelerometer wear, smoking status, alcohol intake, healthy diet score, sleep duration, sleep midpoint, and MVPA accumulated during other two time windows. Additionally, MVPA accumulated in the midday-afternoon period was more protective than in the morning or evening periods, especially with 50–200 min/week of MVPA.

### Timing of MVPA and mortality outcomes

As shown in Table 2, compared with the morning group, both midday-afternoon (model 3: HR = 0.89, 95% CI: 0.81–0.98, P = 0.01) and mixed (model 3: HR = 0.89, 95% CI: 0.80–0.99, P = 0.03) groups showed significantly decreased risks of all-cause mortality. Similarly, midday-afternoon (model 3: HR = 0.72, 95% CI: 0.62–0.84, P = 2.0e−5) and mixed (model 3: HR = 0.74, 95% CI: 0.62–0.88, P = 8.1e−4) groups showed a lower risk of CVD mortality than the morning group. These associations remained statistically significant after multiple testing correction (all FDR-adjusted P < 0.05). However, the evening group showed comparable risks of all-cause mortality and CVD mortality with the morning group, and the four timing groups had a similar risk of cancer mortality (all P > 0.05). Additionally, the survival plots revealed similar findings when comparing standardized mortality risks across four timing groups (Supplementary Figs. 1–3).

Figure 1g–i illustrate the effects of MVPA timing on mortality risk. The associations between MVPA fractions within three time windows and mortality risk were consistent with the main analyses (Table 2). The fractions of MVPA during the three time windows exhibited significant nonlinear associations with mortality outcomes (all $P_{overall} < 0.001$, all $P_{non-linear} < 0.001$), except that the fraction for the evening group was not significantly associated with cancer mortality ($P_{overall} = 0.19$, $P_{non-linear} = 0.07$). More importantly, we observed that the fraction of MVPA during midday-afternoon exhibited lower risks for mortality outcomes than those during morning and evening.

### Sensitivity analyses

Overall, the main results (Table 2) were robust in the comprehensive sensitivity analyses (Supplementary Tables 1–9). The results were robust to sensitivity analyses using different cutoffs (55, 60, 65, and 75%) for assigning groups (Table 3), using the competing risk regression model, using the dataset without imputation, excluding participants with shift work history, excluding participants who wore accelerometers during the daylight-saving time transition, or controlling for the month of accelerometer wear. After additionally controlling for health-related variables, the results for both CVD and cancer mortality were still persistent, although the association between mixed group and all-cause mortality became marginally insignificant (HR = 0.91, 95% CI: 0.82–1.02, P = 0.09). Similarly, when excluding events within the first year of follow-up, censoring up to Dec 31, 2019, or using the subsample with seven-day wear, the results for both CVD mortality and cancer mortality were still consistent, although the association between the mixed group and all-cause mortality became marginally insignificant (P value range: 0.05–0.06).

### Interaction and subgroup analyses

According to their associations with mortality (Table 2), the four timing groups were combined into two groups: midday-afternoon/mixed and morning/evening groups. This combination method facilitates the analysis and interpretation of multiplicative and additive interaction effects, which was widely used in previous studies[24,25]. Synergistic effects of timing of MVPA with age, sex, MVPA level, and CVDs (but not with obesity) on mortality risk were observed (Supplementary

**Table 1 | Baseline characteristics of the study participants**

| Characteristics | Overall (n = 92,139) | Morning (n = 15,865) | Midday-afternoon (n = 41,125) | Evening (n = 8307) | Mixed (n = 26,842) |
|---|---|---|---|---|---|
| **Age at accelerometry (years)** | 62.38 ± 7.84 | 63.54 ± 7.81 | 63.58 ± 7.43 | 59.31 ± 7.96 | 60.81 ± 7.90 |
| **Sex (female/male)** | 52,045/40,094 | 9140/6725 | 22,941/18,184 | 4413/3894 | 15,551/11,291 |
| **White ethnicity** | 89,323 (96.94) | 15,343 (96.71) | 40,183 (97.71) | 7930 (95.46) | 25,867 (96.37) |
| **Townsend deprivation index, median [IQR]** | −2.45 [3.63] | −2.53 [3.51] | −2.53 [3.49] | −2.35 [3.80] | −1.58 [2.88] |
| **Recruitment regions** | | | | | |
| England | 82,716 (89.77) | 14,396 (90.74) | 36,883 (89.69) | 7438 (89.54) | 23,999 (89.41) |
| Wales | 3449 (3.74) | 591 (3.73) | 1538 (3.74) | 340 (4.09) | 980 (3.65) |
| Scotland | 5974 (6.48) | 878 (5.53) | 2704 (6.58) | 529 (6.37) | 1863 (6.94) |
| **Education level** | | | | | |
| Degree or above | 40,348 (43.79) | 6282 (39.62) | 17,732 (43.12) | 3953 (47.59) | 12,377 (46.11) |
| Any other qualification | 44,118 (47.88) | 7943 (50.07) | 19,661 (47.81) | 3935 (47.37) | 12,579 (46.86) |
| No qualification | 7673 (8.33) | 1636 (10.31) | 3732 (9.07) | 419 (5.04) | 1886 (7.03) |
| **Season of accelerometer wear** | | | | | |
| Spring | 20,792 (22.57) | 3476 (21.91) | 9217 (22.41) | 1888 (22.73) | 6211 (23.14) |
| Summer | 24,068 (26.12) | 4017 (25.32) | 9966 (24.23) | 2640 (31.78) | 7445 (27.74) |
| Autumn | 27,583 (29.94) | 4810 (30.32) | 12,700 (30.88) | 2337 (28.13) | 7736 (28.82) |
| Winter | 19,696 (21.38) | 3562 (22.45) | 9242 (22.47) | 1442 (17.36) | 5450 (20.30) |
| **Smoking status** | | | | | |
| Never | 52,952 (57.47) | 8810 (55.53) | 23,330 (56.73) | 4945 (59.53) | 15,867 (59.11) |
| Previous | 33,382 (36.23) | 6130 (38.64) | 15,195 (36.95) | 2761 (33.24) | 9296 (34.63) |
| Current | 5805 (6.30) | 925 (5.83) | 2600 (6.32) | 601 (7.23) | 1679 (6.26) |
| **Alcohol consumption** | | | | | |
| Not current | 5500 (5.97) | 1093 (6.89) | 2377 (5.78) | 456 (5.49) | 1574 (5.86) |
| Two or less times a week | 42,645 (46.28) | 7429 (46.83) | 18,547 (45.10) | 3969 (47.78) | 12,700 (47.31) |
| Three or more times a week | 43,994 (47.75) | 7343 (46.28) | 20,201 (49.12) | 3882 (46.73) | 12,568 (46.82) |
| **Healthy diet score** | 2.69 ± 1.17 | 2.72 ± 1.17 | 2.70 ± 1.17 | 2.61 ± 1.17 | 2.67 ± 1.16 |
| **Sleep duration** | | | | | |
| <7 h/day | 31,750 (34.46) | 5526 (34.83) | 12,651 (30.76) | 3393 (40.85) | 10,180 (37.93) |
| 7–8 h/day | 42,206 (45.81) | 7215 (45.48) | 19,038 (46.29) | 3675 (44.24) | 12,778 (45.74) |
| >8 h/day | 18,183 (19.73) | 3124 (19.69) | 9436 (22.94) | 1239 (14.92) | 4384 (16.33) |
| **Sleep midpoint (hh:mm)** | | | | | |
| <02:30 | 23,823 (25.86) | 5805 (36.59) | 9545 (23.21) | 1769 (21.30) | 6704 (24.98) |
| 02:30-03:30 | 43,025 (46.70) | 7286 (45.92) | 18,934 (46.04) | 3635 (43.76) | 13,170 (49.06) |
| >03:30 | 25,291 (27.45) | 2774 (17.49) | 12,646 (30.75) | 2903 (34.95) | 6968 (25.96) |
| **Health status** | | | | | |
| Obesity | 17,930 (19.46) | 3421 (21.56) | 7829 (19.04) | 1777 (21.39) | 4903 (18.27) |
| Diabetes history | 4265 (4.63) | 872 (5.50) | 1866 (4.54) | 408 (4.91) | 1119 (4.17) |
| Longstanding illness | 27,086 (29.40) | 4917 (30.99) | 12,472 (30.33) | 2312 (27.83) | 7385 (27.51) |
| Depression history | 8171 (8.87) | 1368 (8.62) | 3630 (8.83) | 771 (9.28) | 2402 (8.95) |
| Cardiovascular diseases | 22,661 (24.59) | 4338 (27.34) | 10,702 (26.02) | 1781 (21.44) | 5840 (21.76) |
| Cancer history | 13,428 (14.57) | 2500 (15.76) | 6423 (15.62) | 990 (11.92) | 3515 (13.10) |
| **Total MVPA volume (min/week), median [IQR]** | 113.83 [158.67] | 108.22 [170.00] | 112.83 [153.17] | 104 [148.92] | 120.83 [163.50] |

*IQR* interquartile range, *MVPA* moderate to vigorous physical activity.

Data were mean ± standard deviation or n (%) unless noted otherwise. To assign categories of the timing of MVPA, we divided the clock hours (05:00 to 24:00) into three time windows: morning (05:00–11:00), midday-afternoon (11:00–17:00), and evening (17:00–24:00). These time windows were identified from the exploratory analyses on the timing effects of PA on mortality outcomes (Supplementary Fig. 5). If ≥50% of MVPA occurred during the same time window, participants were assigned to the corresponding timing group. The 50% method we used to define the timing of MVPA is similar to what has been previously used [Qian et al. Diabetes Care. 2021].

Tables 10–14). The beneficial associations of the midday-afternoon/mixed group with all-cause and CVD mortality risks were more prominent among the elderly, less physically active (i.e., below the WHO recommendation) individuals, or those with preexisting CVDs. Additionally, the beneficial association between the midday-afternoon/mixed group and CVD mortality was enhanced among males. Subgroup analyses stratified by age, sex, MVPA level, CVDs, and obesity revealed consistent results with the interaction analyses (Fig. 2 and Supplementary Tables 15–19).

## Discussion
To our knowledge, this large cohort study provides the first evidence that MVPA is associated with lower risks of all-cause, CVD, and cancer mortality regardless of the time of day. Another interesting finding was

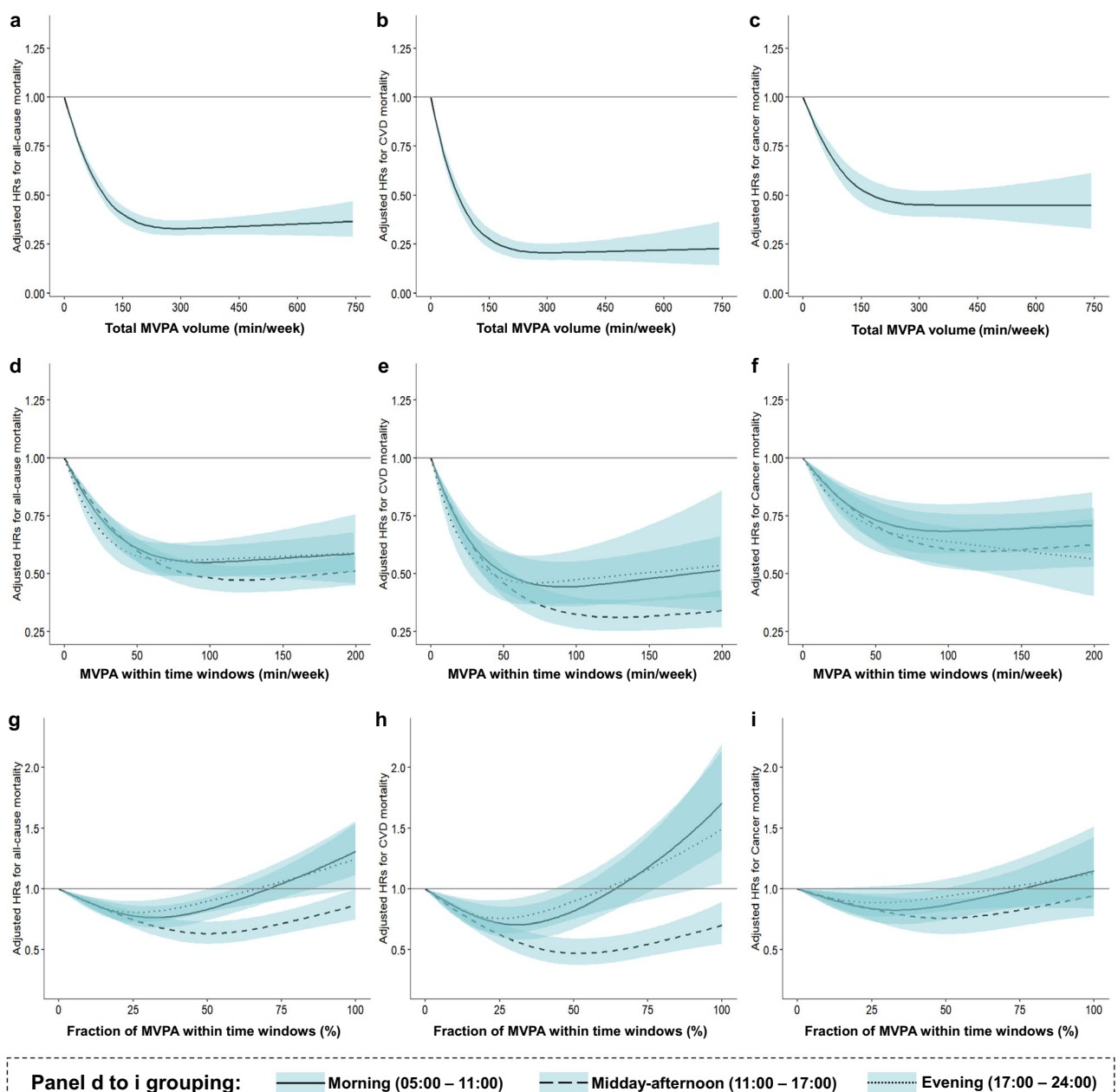

**Fig. 1 | The associations of total MVPA volume, MVPA within time windows, and fractions of MVPA within time windows with mortality risk. a–c** The associations between total MVPA volume and mortality outcomes. The hazard ratios (HRs) were adjusted for age, sex, ethnicity, Townsend deprivation index, recruitment center, education level, the season of accelerometer wear, smoking status, alcohol intake, healthy diet score, sleep duration (<7, 7–8 h, >8 h), and sleep midpoint. **d–f** The associations between MVPA within three time windows and mortality outcomes. The HRs were adjusted for age, sex, ethnicity, Townsend deprivation index, recruitment center, education level, the season of accelerometer wear, smoking status, alcohol intake, healthy diet score, sleep duration (<7, 7–8 h, >8 h), sleep midpoint, and MVPA volume during other two time windows. **g–i** The associations between the fractions of MVPA within three time windows with mortality risk (indicating MVPA timing effects). The HRs were adjusted for age, sex, ethnicity, Townsend deprivation index, recruitment center, education level, the season of accelerometer wear, healthy diet score, smoking status, alcohol intake, sleep duration (<7, 7–8 h, >8 h), sleep midpoint, and total MVPA volume. CVD cardiovascular disease, MVPA moderate to vigorous physical activity. Error bands represent the 95% confidence intervals for each effect estimate. Source data are provided as a Source Data file.

that the midday-afternoon and mixed MVPA timing groups, as compared to the morning group, showed substantially decreased all-cause and CVD mortality risks, but not cancer mortality. The associations between MVPA timing and mortality risk were independent of socio-demographic factors, lifestyle, comorbidities, sleep duration, sleep midpoint, and total MVPA volume. These findings were robust to multiple testing corrections and sensitivity analyses. In addition, the observed protective effects of MVPA timing were more pronounced among the elderly, males, less active individuals, or those with preexisting CVDs.

Previous human trials have mainly focused on the short-term effects of PA timing on metabolic functions and yielded inclusive and contradictory findings[16–18]. The vast majority of these studies only used structured exercise to investigate the health effects of MVPA timing[16,17]. However, not all MVPA occurs during structured physical exercise, which may cause confounding bias, thereby potentially contributing to the large discrepancies in the previous evidence. In contrast, the current study captured the full spectrum of objectively measured MVPA in free-living settings[20], which is vital for defining the MVPA phenotype. To date, only a few studies using accelerometers have

**Table 2 | The associations between the timing of MVPA and mortality risk**

| Outcomes | Events/n | Person-years | Model 1 HR (95% CI); P | Model 2 HR (95% CI); P | Model 3 HR (95% CI); Pª |
|---|---|---|---|---|---|
| **All-cause mortality** | 3088/92,139 | 638,825 | | | |
| Morning | 652/15,865 | 109,537 | 1.00 (reference) | 1.00 (reference) | 1.00 (reference) |
| Midday-afternoon | 1432/41,125 | 284,481 | 0.85 (0.78–0.94); 7.5e-4 | 0.87 (0.79–0.96); 0.003 | 0.89 (0.81–0.98); 0.01ª |
| Evening | 249/8307 | 57,768 | 1.05 (0.91–1.21); 0.54 | 1.05 (0.91–1.22); 0.49 | 0.98 (0.84–1.13); 0.76 |
| Mixed | 755/26,842 | 187,039 | 0.88 (0.79–0.97); 0.01 | 0.89 (0.80–0.99); 0.03 | 0.89 (0.80–0.99); 0.03ª |
| **CVD mortality** | 1076/92,139 | 638,825 | | | |
| Morning | 274/15,865 | 109,537 | 1.00 (reference) | 1.00 (reference) | 1.00 (reference) |
| Midday-afternoon | 464/41,125 | 284,481 | 0.66 (0.57–0.77); 4.4e-8 | 0.68 (0.59–0.79); 6.8e-7 | 0.72 (0.62–0.84); 2.0e-5ª |
| Evening | 90/8307 | 57,768 | 0.94 (0.74–1.20); 0.64 | 0.96 (0.76–1.22); 0.74 | 0.87 (0.68–1.11); 0.26 |
| Mixed | 248/26,842 | 187,039 | 0.72 (0.60–0.85); 1.7e-4 | 0.73 (0.62–0.87); 4.9e-4 | 0.74 (0.62–0.88); 8.1e-4ª |
| **Cancer mortality** | 1872/92,139 | 638,825 | | | |
| Morning | 362/15,865 | 109,537 | 1.00 (reference) | 1.00 (reference) | 1.00 (reference) |
| Midday-afternoon | 888/41,125 | 284,481 | 0.95 (0.84–1.08); 0.42 | 0.97 (0.85–1.09); 0.58 | 0.97 (0.86–1.10); 0.64 |
| Evening | 154/8307 | 57,768 | 1.14 (0.94–1.37); 0.19 | 1.14 (0.94–1.38); 0.18 | 1.07 (0.89–1.30); 0.47 |
| Mixed | 468/26,842 | 187,039 | 0.96 (0.83–1.10); 0.53 | 0.96 (0.84–1.11); 0.61 | 0.96 (0.84–1.11); 0.60 |

Cox proportional hazard regression was used to examine the associations. Model 1 was adjusted for age and sex. Model 2 was adjusted as in model 1 and for ethnicity, Townsend deprivation index, recruitment center, education level, the season of accelerometer wear, healthy diet score, smoking status, and alcohol intake. Model 3 was adjusted as in model 2 and for sleep duration (<7, 7–8, >8 h), sleep midpoint, and total MVPA volumes.

CVD cardiovascular disease, HR hazard ratio, MVPA moderate to vigorous physical activity.

ªP values remained significant after multiple testing with the FDR method.

**Table 3 | Sensitivity analysis on the associations between timing of MVPA and mortality risk by using different cutoffs for timing group assignmentª**

| Outcomes | Cutoff = 55% HR (95% CI); Pᵇ | Cutoff = 60% HR (95% CI); Pᵇ | Cutoff = 65% HR (95% CI); Pᵇ | Cutoff = 70% HR (95% CI); Pᵇ |
|---|---|---|---|---|
| **All-cause mortality** | | | | |
| Morning | 1.00 (reference) | 1.00 (reference) | 1.00 (reference) | 1.00 (reference) |
| Midday-afternoon | 0.86 (0.77–0.95); 0.003ª | 0.84 (0.75–0.95); 0.004ª | 0.80 (0.70–0.90); 4.6e-4ª | 0.82 (0.71–0.95); 0.01ª |
| Evening | 0.95 (0.80–1.13); 0.58 | 0.90 (0.74–1.09); 0.28 | 0.95 (0.76–1.18); 0.62 | 1.02 (0.80–1.31); 0.87 |
| Mixed | 0.82 (0.74–0.91); 1.5e-4ª | 0.80 (0.72–0.89); 6.9e-5ª | 0.74 (0.66–0.83); 6.2e-7ª | 0.75 (0.66–0.86); 2.6e-5ª |
| **CVD mortality** | | | | |
| Morning | 1.00 (reference) | 1.00 (reference) | 1.00 (reference) | 1.00 (reference) |
| Midday-afternoon | 0.72 (0.61–0.85); 8.3e-5ª | 0.71 (0.59–0.86); 2.9e-4ª | 0.66 (0.54–0.81); 6.5e-5ª | 0.70 (0.55–0.89); 0.004ª |
| Evening | 0.87 (0.66–1.14); 0.32 | 0.81 (0.59–1.11); 0.19 | 0.86 (0.60–1.22); 0.39 | 0.93 (0.62–1.40); 0.72 |
| Mixed | 0.67 (0.57–0.80); 3.9e-6ª | 0.67 (0.56–0.80); 5.0e-6ª | 0.62 (0.52–0.74); 2.4e-7ª | 0.68 (0.55–0.83); 2.4e-4ª |
| **Cancer mortality** | | | | |
| Morning | 1.00 (reference) | 1.00 (reference) | 1.00 (reference) | 1.00 (reference) |
| Midday-afternoon | 0.95 (0.82–1.09); 0.43 | 0.99 (0.84–1.15); 0.86 | 0.87 (0.74–1.04); 0.13 | 0.88 (0.72–1.08); 0.22 |
| Evening | 1.04 (0.83–1.30); 0.72 | 1.06 (0.82–1.37); 0.65 | 1.04 (0.78–1.39); 0.77 | 1.04 (0.75–1.46); 0.80 |
| Mixed | 0.93 (0.81–1.07); 0.32 | 0.95 (0.82–1.11); 0.54 | 0.83 (0.71–0.97); 0.02 | 0.81 (0.68–0.97); 0.02 |

Cox proportional hazard regression was used to examine the associations, which were adjusted for age, sex, ethnicity, Townsend deprivation index, recruitment center, education level, the season of accelerometer wear, healthy diet score, smoking status, alcohol intake, sleep duration (<7, 7–8, >8 h), sleep midpoint, and total MVPA volume (Model 3).

CVD cardiovascular disease, HR hazard ratio, MVPA moderate to vigorous physical activity.

ªWe did not conduct sensitivity analyses by using the cutoffs of greater than 70% due to the small sample size for timing groups (other than mixed groups) by doing that.

ᵇP values remained significant after multiple testing with the FDR method.

explored the long-term health effects of MVPA timing and they were conducted among patients with type 2 diabetes[20,26]. A cross-sectional study found that the morning MVPA timing group had the highest CVD risk score among males with type 2 diabetes[20]. A cohort study based on patients with type 2 diabetes found no significant difference in all-cause mortality risk between the before and after 13:00 MVPA time groups[26]. These previous studies defined time windows using equally spaced intervals (which do not appear to be clinically driven[27]) or in an arbitrary manner. Thus, to better evaluate the association between MVPA timing and health, the identification of time windows in the current study was driven by exploratory analyses of the exposure-outcome relationships.

We speculate some possible reasons that may account for the beneficial effects of MVPA timing. One possible reason could be related to the circadian modulation of cardiometabolic reactivity to physical exercise. For instance, the unfavorable timing of MVPA we found is very close to the morning (~6:00–12:00) and evening (~18:00–22:00) peaks for CVD risk[28,29]. In line with this, human trials found time-of-day specific cardiovascular reactivity to physical exercises, such as the greatest vagal withdrawal at ~9:00[30], peaks of catecholamine reactivity at ~9:00 and ~21:00[30], and faster recovery of systolic blood pressure after exercise in the late afternoon (~17:00) than in the early morning (~8:30)[31]. In addition, this explanation may also be supported by our findings of stronger associations between unfavorable timing of MVPA

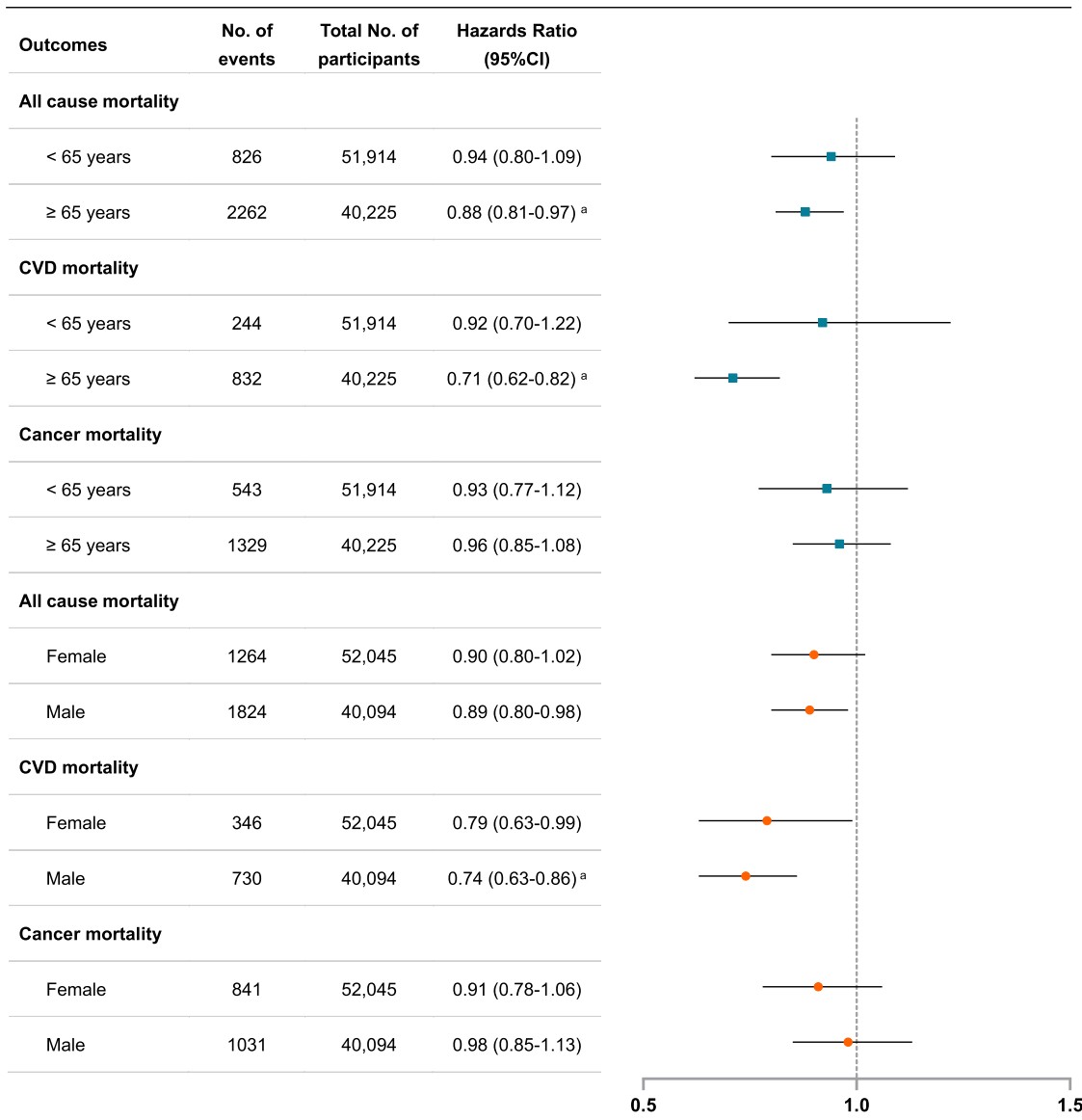

**Fig. 2 | Subgroup analysis on the associations between favorable timing of MVPA (midday-afternoon & mixed vs. morning & evening) and mortality risk stratified by age and sex categories.** [a]*P* values remained significant after multiple testing with the FDR method. Cox proportional hazard regression was used to examine the associations, which were adjusted for age, sex, ethnicity, Townsend deprivation index, recruitment center, education level, the season of accelerometer wear, smoking status, alcohol intake, healthy diet score, sleep duration (<7, 7–8, >8 h), sleep midpoint, and total MVPA volume (Model 3). According to their associations with mortality (Table 2), four timing groups were combined into two groups: (1) favorable timing of MVPA (i.e., midday-afternoon or mixed); (2) unfavorable (i.e., morning or evening). This combination method facilitated the analysis and interpretation of multiplicative and additive interaction effects, which was widely used in previous studies [Shan et al. *BMJ*, 2018.; Huang et al. *Br J Sports Med*. 2021]. CVD cardiovascular disease, CI confidence interval, MVPA moderate to vigorous physical activity. Error bars represent the 95% CIs for each effect estimate. Source data are provided as a Source Data file.

and higher mortality risk among individuals with preexisting CVDs. Another possible explanation is that MVPA timing could be linked to circadian variations in other behavioral/environmental factors (e.g., eating and light exposure). For example, post-meal walking has been shown to be more effective than morning/afternoon walking in improving glucose control[18]. Another possibility is that the potential differences in exercise mode across timing groups may contribute to the associations between MVPA timing and health risk. It has been reported that combined training can cause a greater decrease in body fat than aerobic or resistance training alone[32]. However, time-of-day specific data of both PA mode and behavioral/environmental factors were not available in this study, and we were not able to test these hypotheses. Thus, the underlying mechanisms remain to be elucidated.

Another interesting finding of this present study is that age, sex, MVPA levels, and preexisting CVDs may modify the associations between MVPA timing and mortality risk, particularly CVD mortality. A similar synergistic effect between sex and MVPA timing has been reported in a previous cross-sectional study of patients with type 2 diabetes in which the association between morning MVPA and a higher coronary heart disease risk score was only observed in men[20]. To the best of our knowledge, we also report an interesting result regarding the interaction between age and MVPA timing. The possible reasons might be related to sex-specific and age-specific physiological responses to PA[33,34]. There may be a concern that potential variations in total MVPA across age and sex subgroups may contribute to the observed interaction effects, but further adjustment for total MVPA did not change the results, suggesting that total MVPA is unlikely to

contribute to age and sex effects. In addition, the finding of synergistic effects of MVPA timing with MVPA levels and preexisting CVDs also extends our knowledge in this field. Further studies are needed to explore the mechanisms underlying these synergistic effects.

Our findings have several implications. First, the results of this study suggest that the timing of MVPA may have an impact on health. A better understanding of how different dimensions of PA affect the body can help to maximize its health benefits. It should also be emphasized that MVPA in any time windows was associated with lower risks of mortality outcomes investigated. Combining these two findings should help guide and improve strategy-making and practice in PA-related health areas. Second, the timing of MVPA can be easily measured objectively and is highly reproducible due to the development of wearable devices and technology[6]. The accessibility and convenience of accelerometer measurement make the impact of MVPA timing and measurement of MVPA more meaningful and actionable in public health. Third, this study may help to identify individuals (i.e., the elderly, males, less active individuals, and those with preexisting CVDs) who could have greater beneficial effects of choosing a more favorable MVPA timing. This may also provide insights for planning more effective intervention strategies. Finally, our findings highlight the importance of incorporating circadian rhythms in future PA-related research.

The strengths of this study include its population-based sample, prospective design, objective measurement of MVPA in a free-living setting, careful control of covariates, and comprehensive sensitivity and interaction analyses. However, this study has some limitations. First, the UK Biobank sample is healthier and lives in less socio-economically deprived areas than the general population. Although this sample is not suitable for estimating the prevalence and incidence of health conditions, valid estimates of exposure-outcome relationships are widely generalizable[35]. Second, the use of accelerometry to measure PA has its limitations: PA is measured in absolute terms and a certain absolute volume of PA may be different for each individual, depending on their fitness level. The single accelerometer was not able to detect all fine details of PA, such as upper/lower body movements. Third, although 7-day monitoring periods have been routinely used in PA monitor studies because they provide the opportunity to sample PA on both week and weekend days and achieve intra-class correlations of greater than 80% in most populations[36], it is still unclear whether a 7-day accelerometer measurement is representative of longer-term behaviors[37], especially for PA timing. In addition, although our findings were robust to daylight-saving time and adjustment of the season (even month) of accelerometer wear, these sensitivity analyses may not be sufficient to account for potential misclassification or seasonal variation of PA timing. These situations highlight the importance of repeated and long-term administration of accelerometers in future studies. Fourth, despite the inclusion of a wide range of confounders in the analyses, residual or unmeasured confounding cannot be ruled out in our study, such as employment status, mode of PA, and meal timing. The observational nature of this study precludes conclusions about definitive causal relationships. Finally, some covariates such as lifestyles were mainly collected during the physical visits to the assessment centers, nearly 5.6 years before the present study baseline. However, since these covariates are generally stable over time[6], this time lag is unlikely to weaken our findings.

In summary, our results show that MVPA timing may have the potential to maximize the health benefits of daily PA. Future experimental studies in humans are needed to confirm the impact of the timing of various types of PA on subsequent health outcomes.

## Methods

### Participants and accelerometer assessment

This large cohort study was conducted based on the UK Biobank. The UK Biobank received ethical approval from the North West Multi-center Research Ethics Committee to collect and distribute samples and data from the participants (Reference numbers: 16/NW/0274 & 21/NW/0157; https://www.ukbiobank.ac.uk/learn-more-about-uk-biobank/governance/ethics-advisory-committee), which covers the work in this study under approved Application 58082. All UK Biobank participants provided informed consent. In addition, we have obtained approvals from our institutions for analyzing the UK Biobank data in this study. UK Biobank is a large population-based cohort that recruited over 500,000 participants aged 40–73 years between 2006 and 2010[38]. Participants visited one of 22 assessment centers across England, Scotland, and Wales, and underwent detailed baseline assessments, including various sociodemographic, lifestyle, health, and physical assessments. Details of the rationale, design, and measurements for the UK Biobank are available online (www.ukbiobank.ac.uk). Between February 2013 and December 2015 (on average, approximately 5.5 years after their initial baseline recruitment), 236,519 UK Biobank participants were invited to participate in an accelerometer study. Among them, 106,053 participants agreed to participate and were provided with a wrist-worn accelerometer (Axivity AX3)[39]. Participants who accepted accelerometry measurement showed similar baseline demographic and health-related characteristics as those who declined the measurement[40]. The accelerometer was set up to start at 10 a.m. two working days after postal dispatch (to ensure that the accelerometer would not start recording during delivery), and capture triaxial acceleration data over 7 days at 100 Hz with a dynamic range of ±8 gravity. Participants were instructed to wear the device on their dominant wrist continuously for seven days while continuing with their usual activities. Participants were asked to mail the device in a pre-paid envelope back to the coordinating centers, after the seven-day monitoring period.

Using the raw accelerometer data from 103,682 participants, the UK Biobank accelerometer expert working group conducted data processing and generated physical activity intensity data (average vector magnitude in milligravity units) in 5-s epochs (field ID 90004) using the raw accelerometer data (field ID 90001). The raw acceleration signals were calibrated to gravity. Non-wear time was defined as consecutive stationary episodes lasting for at least one hour where all three axes had a standard deviation of less than 13.0 milligravity[39]. Epochs representing non-wear time were imputed based on all wear-time data at a similar time of the day on different days for each participant. More details about the data processing and analysis have been published[39].

The exclusion criteria are as follows: (1) those who withdrew from UK Biobank; (2) those who had no PA data in any one hour of the 24-h cycle; (3) Similar to the previous study[20], those who had high nocturnal activity (>10% PA accumulated between 01:00 and 04:00), as we focused on individuals with a diurnal lifestyle; (4) those with unreliable or invalid accelerometry data. The criteria for unreliable or invalid accelerometry data included (1) unexpectedly small or large size (Field ID: 90002); (2) less than 72 h or did not provide data for all 1-h periods within a 24-h cycle during the 7-day data collection (Field ID: 90015); (3) not well-calibrated (Field ID: 90016); (4) recalibrated using the previous accelerometer record from the same device worn by a different participant (Field ID: 90017); (5) data with a non-zero count of interrupted recording periods (Field ID: 90180); (6) data with more than 768 (Q3 + 1.5 × IQR) data recording errors (Field ID: 90182). In total 11,543 participants were excluded. Finally, 92,139 participants (88.87%) with valid data were included in the current study for the main analysis with the imputation of missing data, while 89,141 participants with the complete set of data were included for sensitivity analysis (Supplementary Fig. 4).

### Exposure

The processed activity intensity data were further used to yield MVPA. MVPA, often defined as requiring a moderate to a large amount of

effort and with a notable to substantial acceleration in heart rate, is a well-validated surrogate for PA[40]. More importantly, as suggested by the previous research[20], focusing on high-intensity levels of PA, such as MVPA, helps to determine a clear timing effect. Light-intensity PA was not included in this study because it occurs during walking and even sitting hours, thereby obscuring the temporal distribution of the more effective PA with higher intensity[20]. In this study, we tried to generate PA timing grouping using the averaged acceleration data, and only 0.74% ($n = 680$), 10.1% ($n = 9316$), and 0.70% ($n = 641$) of the participants were assigned to the morning, midday-afternoon, and evening group, respectively. The remaining 88.5% ($n = 81,502$) of the participants were assigned to the mixed group. Therefore, similar to previous research[20], we focused on PA at a relatively high-intensity level (i.e., MVPA) to determine a robust timing phenotype.

Moderate-intensity physical activity was collected in sessions (5-min periods where more than 80% of 5-s epochs had a mean acceleration of 100 to 400 milligravity)[40]. Vigorous-intensity physical activity was defined as the 5-s epochs, where the mean acceleration was above 400 milligravity[41]. We included individuals with a diurnal lifestyle. Furthermore, high nocturnal activity always means sleep disturbances. Due to these reasons, we calculated the total minutes of MVPA by summing the minutes of moderate-intensity physical activity and vigorous-intensity physical activity between 05:00 and 24:00. Among 8354 (9.07%) participants who provided <1 week of accelerometry data, we extrapolated MPVA data to seven days.

Then, we ran the exploratory analyses to determine the appropriate boundaries for the time windows used for the main analyses. Exposure-dependent methods, such as the equally spaced intervals (which do not appear to be clinically driven), are generally arbitrary and may not be helpful in assessing a variable's actual predictive value[27]. In contrast, the outcome-based methods allow an "optimal" cutoff to be estimated[42,43]. Therefore, this study ran an exploratory analysis for the identification of time window boundaries using the 'outcome-based' method. In addition, to balance sample size and accuracy, we chose the 2-h time window intervals (3-h only for the 21:00–24:00 period) in the exploratory analyses. The 50% method of assigning timing groups was similar to that previously used[20], and this method avoids the participants being assigned to multiple timing groups. If over 50% of total daily MVPA occurred during the same 2-h period, participants would be assigned to the corresponding groups. For those who spent less than 50% of the total daily MVPA in any of the 2-h time windows, we assigned them to the mixed group. As shown in Supplementary Fig. 5, which mimics the nonlinear exposure-outcome curves, two change points at 11:00 and 17:00 were consistently observed for all mortality outcomes. Compared with the mixed group, the 2-h timing groups of the morning (05:00–11:00) and evening (17:00–24:00) periods seemed to have higher mortality risks. The 2-h timing groups of the midday to afternoon period (11:00–17:00) presented comparable mortality risks with the mixed group (Supplementary Fig. 5). Finally, morning (05:00–11:00), midday to afternoon (11:00–17:00), and evening (17:00–24:00) time windows were used in subsequent MVPA timing grouping and statistical analyses.

Similar to the previous study[20], if ≥50% of total daily MVPA occurred during the same time window, participants would be assigned to the corresponding MVPA timing groups: morning (05:00–11:00), midday-afternoon (11:00–17:00), and evening (17:00-24:00) groups. For those who spent <50% of the total daily MVPA in any of three time windows, we assigned them to the mixed group.

## Outcomes

The outcomes were all-cause, CVD, and cancer mortality. Cause-specific mortality was ascertained using the International Statistical Classification of Diseases and Related Health Problems, Tenth Revision (Supplementary Table 20). We measured specific mortality due to CVDs (codes I00–I99) and cancer (codes C00-C97) using the death

registry. The date and cause of death were obtained from the death datasets of the National Health Service Information Center and the NHS Central Register. At the time of analysis, we censored the Cox regression analyses at the date of death or the date of available mortality data (12 November 2021), whichever came first.

## Covariates

Data on these possible covariates were obtained using self-reported questionnaires, accelerometers, and registry records. Age (continuous) was calculated from the date of birth and the date of wearing the accelerometer. Self-reported questionnaires were used to determine sex (female/male), ethnicity (white/others), recruitment center (England/Wales/Scotland), and Townsend deprivation index (continuous) based on the postcode of residence using aggregated data on unemployment, car and home ownership, and household overcrowding. The data on the season of accelerometry wear (i.e., spring, March to May; summer, June to August; autumn, September to November; winter, for December to February; UK Meteorological Office definitions) was obtained from accelerometer data. Other covariate data including educational attainment (degree or above/any other qualification/no qualification), smoking status (never/previous/current), frequency of alcohol intake (not current/less than three times a week/three or more times a week), and diet-related factors were obtained from touchscreen questions. We calculated the healthy diet score by using the following factors: vegetable intake of at least four tablespoons each day (median), fruit intake of at least three pieces each day (median), fish intake of at least twice per week (median), unprocessed red meat intake of no more than twice per week (median), and processed meat intake of no more than two per week (median). Sleep duration (<7 h per day/7–8 h per day/> 8 h per day) and sleep midpoint (<02:30/02:30-03:30/> 03:30) were measured using the accelerometer. Obesity (body mass index ≥30 kg/m²) was obtained from touchscreen questions. Previous diagnoses of diabetes, longstanding illness, depression, CVDs, and cancer were obtained from the self-reported questionnaires, hospital records, and death registry. The values of some covariates, including education level, smoking status, alcohol consumption, healthy diet score, obesity, diabetes history, longstanding illness, and cancer history, were obtained from touchscreen questionnaires at the time-point closest to the accelerometry (Supplementary Fig. 6). Detailed information sources, assessment timeline, and missing percentages are shown in Supplementary Fig. 6 and Supplementary Tables 20–21.

## Statistical analyses

The event numbers of all outcomes were sufficient as per the rule-of-thumb estimation[44], which requires at least ten events per variable. We conducted multiple imputations to assign any missing covariate values using the "mice" package (v3.13.0) in R[45]. The overall sample and complete case sample showed similar baseline characteristics (Supplementary Table 22). Before investigating the timing effect of MVPA, the linear and nonlinear associations of total MVPA volume and MVPA within the three time windows with mortality risk were assessed using penalized cubic splines fitted in the fully adjusted Cox models. In addition, we assessed the linear and nonlinear associations between the proportions of MVPA accumulated within the three time windows and mortality risk. Based on the fully adjusted model, cumulative risk curves were generated to show the standardized risks of mortality outcomes according to MVPA timing groups. Collinearity between all covariates was examined via correlation matrix analysis, which revealed no problem of multicollinearity. Cox proportional hazard regression (using the "survival" package v3.2-11 in R) was used to examine the associations of the timing of MVPA with mortality. HRs and their 95% CIs were calculated. We conducted careful adjustments. Model 1 adjusted for age and sex. Model 2 additionally adjusted for ethnicity, Townsend deprivation index, recruitment center, education

level, the season of accelerometer wear, smoking status, alcohol intake, and healthy diet score. To investigate the associations independent of sleep duration, sleep phase, and total MVPA volume, model 3 further adjusted for these factors.

We performed a series of sensitivity analyses. First, we used different MVPA fraction cutoffs (55, 60, 65, and 70%) to assign timing groups. We did not conduct sensitivity analyses using cutoffs of >70% due to the small sample size for timing groups (other than mixed group). Second, Fine-Gray subdistribution hazards (using the "cmprsk" package v2.2-10 in R) were calculated, incorporating other-cause death as a competing risk for cause-specific mortality[46]. Third, we further adjusted for health-related variables potentially on causal pathways[6], including obesity, diabetes history, longstanding illness, depression history, CVDs, and cancer. Fourth, we restricted the analyses to participants without shift work history and those without any missing covariate data, respectively. Fifth, we excluded participants who wore accelerometers during daylight-saving time transitions. Sixth, we ran the analyses by controlling for the month of accelerometer wear instead of the season of accelerometer wear. Seventh, we excluded events that occurred within one year of follow-up. In addition, we performed analyses by censoring up to 31 Dec 2019 (the start of the COVID-19 pandemic[47]). Finally, we repeated the analyses among those with ≥6 days of accelerometer wear.

Multiplicative and additive interaction (using the "interactionR" package v0.1.3.9000 in R) analyses and subgroup analyses were performed on age, sex, MVPA level (meeting the WHO recommendation[1,3] or not), CVDs, and obesity (BMI ≥30 kg/m$^2$). All statistical tests were two-sided, and a $P$ value of <0.05 was regarded as statistically significant. To account for multiple testing, $P$ values in fully adjusted models were corrected using the false-discovery rate (FDR)[48]. All statistical analyses were performed using R v4.0.4 and SPSS v26. R codes are available with the online version of this article (Supplementary Code) and at https://github.com/hlfeng99/Supplementary-Code.

### Reporting summary
Further information on research design is available in the Nature Portfolio Reporting Summary linked to this article.

## Data availability
The UK Biobank data were available from the UK Biobank and can be accessed by researchers on the application (www.ukbiobank.ac.uk/). Source data are provided with this paper.

## Code availability
The simulation code is available in the online version of this article.

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

## Acknowledgements

This work was supported by the National Key R&D Program of China (2021YFC2501500, J.Z.), the National Natural Science Foundation of China (82171476, J.Z. and 82101558, H.F.), and the China Postdoctoral Science Foundation (2021T140139, H.F.; 2021TQ0081, Y.L.; and 2021M690750, Y.L.). H.F., Y.L., and J.W. were supported by the International Postdoctoral Exchange Fellowship Program of China. The funders had no role in the design and conduct of the study; collection, management, analysis, and interpretation of the data; preparation, review, or approval of the manuscript; and decision to submit the manuscript for publication. This research has been conducted using the UK Biobank Resource under Application Number 58082. We thank the participants of the UK Biobank.

## Author contributions

All authors meet authorship criteria and no others meeting the criteria have been omitted. All authors reviewed and approved the manuscript. H.F., L.Y., and Y.Y.L. contributed to the statistical analyses, data interpretation, and had the primary responsibility for writing the manuscript, with the help of S.A., Y.P.L., Y.L., X.J., B.L., and J.W. The revision of the manuscript was conducted by S.A., N.Z., N.Y.C., X.T., C.B., X.C., J.W.Y.C., R.K.W.S., and Y.K.W. In addition, J.Z., Y.K.W., and H.F. contributed to the conception and design of the study. H.F., L.Y., Y.Y.L., and J.Z. had full access to all the data in the study and took responsibility for the integrity of the data and the accuracy of the data analysis. Y.K.W. is the senior author.

## Competing interests

The authors declare no competing interests.
