## [Peer Review File · Nature Communications]

Reviewers' Comments:

Reviewer #1:

Remarks to the Author:

I read the manuscript about associations of physical activity with all-cause and cause-specific mortality with great interest. There is little prior research on the question whether exercise/physical activity early or late in the day gives most health benefits, so this study represents noteworthy results. It is good that the authors underscore the benefits of PA in terms of survival, independent of time of day.

Overall, the methods are sound. The use of accelerometry to measure physical activity has its limitations: the activity is measured in absolute terms and a certain (absolute) volume of activity will be different for each individual, depending of their fitness level. This should be included as a limitation and elaborated upon in the discussion.

It is important that the benefit of PA in terms of reduced mortality gets through (irrespective of time of day). Would it be possible, based on your data, to say something about the difference in dose compared to the difference in time-of-day? It might be very important in a public health perspective (and message to people) that those who habitually are active in the morning or late evening don't get the impression that it is not good for them (even if other times of the day may be even better).

It is good that the authors have teased out differences between age groups and sexes. Would it be possible to do the same for BMI categories (or other measures of cardiometabolic health)? There might be differences in response which are dependent on these factors.

Page 10, lines 236-237: I suggest modifying your statement about the implications of the findings: I will say that these data suggest that timing of MVPA may have impact on health (based on the fact that these are observational data and there might be other confounding variables that you did not adjust for). There is clearly some confounding present as the estimates are less clear the more you adjust in your models. Do you for example have employment status of your participants?

Minor comments:

I would rephrase the second sentence of the introduction (replacing suggested with suggest).

In the results, I would move the total number of deaths etc (now on page 7, second paragraph) up, as this is important information to have in mind when reading all the results.

In the discussion, I do not see the relevance of bringing in the findings from reference 37.

It is unclear to me what 'it' refers to in line 242 on page 10.

Reviewer #2:

Remarks to the Author:

General Comments: This is a well documented secondary analysis of the UK Biobank database seeking to examine the association of moderate to vigorous physical activity (MVPA) with total, cardiovascular and selected cancer mortality. The uniqueness of this study is the examination of objectively measured MVPA (wearables/accelerometry) among 92K+ adults and specifically whether a daily time period (morning, mid-day, and evening) vary among these associations. The methods are well described, use of an excellent and unique data source, and clear study rationale supported by a thorough literature review. The analytic approach is most appropriate in using Cox proportional hazards modeling and expressing the data using person years with calculated hazard ratios and 95% CIs. The results are well presented complemented by both narrative and tabular data. Supplementary figures and tables provide further insights into the results with a review of

the sensitivity analyses. The discussion is concise and stays within the bounds of the data presented. Further explanation in the limitations section may shed further light on the MVPA volume differences that exist between the morning (referent) and mid-day and evening groups. Where the morning group presents with a visibly lower volume of MVPA than the mid-day and mixed groups, with the evening group showing the lowest volume.

Did the authors seek to stratify by volume of MVPA across each of the time groups -- hence examining each time period controlling for volume, or did I miss this adjustment.

The most significant and satisfying finding is that regardless of time of day, those participating in significant doses of MVPA accrued a protective effect from pre-mature mortality.

Reviewer #3:

Remarks to the Author:

This was a generally well-written report on an interesting and important investigation using UK Biobank data. The authors performed a thorough analysis of the relationship between objectively measured MVPA timing and all-cause, CVD, and cancer mortality. This addresses a substantial gap in current knowledge. The manuscript was enhanced by thoughtful sensitivity and supplemental analyses. I do, however, have some comments and questions for the authors, as listed below.

-Lines 73-75 – Supplementary figure 2 is referred to in support of the time windows. Were subjects required to accumulate 50+% of their MVPA in a 2-hour period in order to be classified in one of the non-mixed categories? This post-hoc approach would seem to make it more likely to generate groups with significant differences.

-Line 89 – Analyses were adjusted for season of accelerometer wear. However, this does not seem sufficient to account for potential misclassification of PA timing due to season of measurement.

-Lines 91-92. Based on Supplementary Figure 3, approximately 30,000 participants had a second study interview – which may have occurred before or after the accelerometry measurement period. Can the authors clarify what information was collected at these subsequent interviews and how they were used in the analysis. For example, did some subjects have time-varying covariates entered in the models?

-Lines 107-108. Did assessment of health-related variables occur only at the recruitment interview or at other times. If done at the recruitment interview, might these pre-existing conditions influence the timing and amount of MVPA during the later accelerometer wear period? If so, how would that influence the interpretation of the models which include these health-related variables?

-Lines 112-113. In the main analysis, was there a minimum number of accelerometer wear days required for inclusion? How was a valid accelerometer wear day defined?

-Line 121. I think you mean “process” instead of “progress”.

-Figure 1. In the legend, both the morning and evening times appear to be represented by solid lines (darker for morning). However, there is only one solid line in the graphs. Please adjust the legend to clarify.

-Figure 1. Please confirm that graphs G-I were additionally adjusted for only total MVPA volume (compared to graphs D-F). This is what is stated in the figure title but in lines 101-102 it is stated that the third model is additionally adjusted for sleep duration, sleep phase, as well as total MVPA volume (and sleep duration and sleep phase are listed in model 2 for graphs D-F).

-Line 225. I think you mean “test” instead of “testify”.

-Line 245. Regarding potential limitations, how robust is your estimate of MVPA timing? Is 1-week behavior reflective of longer-term behavior? If not, how might this influence the interpretation of these results based on only 7-days of data?

****Responses to the Reviewer #1****

Comment/Question 1: I read the manuscript about associations of physical activity with all-cause and cause-specific mortality with great interest. There is little prior research on the question whether exercise/physical activity early or late in the day gives most health benefits, so this study represents noteworthy results. It is good that the authors underscore the benefits of PA in terms of survival, independent of time of day.

Response 1: We appreciate the reviewer's assessment and positive comments.

Comment/Question 2: Overall, the methods are sound. The use of accelerometry to measure physical activity has its limitations: the activity is measured in absolute terms and a certain (absolute) volume of activity will be different for each individual, depending of their fitness level. This should be included as a limitation and elaborated upon in the discussion.

Response 2: We thank the reviewer for pointing out this issue. Accordingly, we have revised the Discussion section to clarify the limitations of the use of accelerometry to measure physical activity:

“Second, the use of accelerometry to measure PA has its limitations: PA is measured in absolute terms and a certain (absolute) volume of PA may be different for each individual, depending on their fitness level. The single accelerometer was not able to detect all fine details of PA, such as upper/lower body movements. Third, although 7-day monitoring periods have been routinely used in PA monitor studies because they provide the opportunity to sample PA on both week and weekend days and achieve intra-class correlations of greater than 80% in most populations³⁵, it is still unclear whether a 7-day accelerometer measurement is representative of longer-term behaviors³⁶, especially for PA timing. In addition, although our findings were robust to daylight saving time and adjustment of season (even month) of accelerometer wear, these sensitivity analyses may not be sufficient to account for potential misclassification or seasonal variation of PA timing. These situations highlight the importance of repeated and long-term administration of accelerometers in future studies.” [Discussion (page 8; line 183-193)]

Comment/Question 3: It is important that the benefit of PA in terms of reduced mortality gets through (irrespective of time of day). Would it be possible, based on your data, to say something about the difference in dose compared to the difference in time-of-day? It might be very important in a public health

perspective (and message to people) that those who habitually are active in the morning or late evening don't get the impression that it is not good for them (even if other times of the day may be even better).

Response 3: We fully agree with the reviewer that an accurate message of the beneficial health effects of MVPA is of major public interest. In the previous version, we showed that MVPA minutes within three time windows were all significantly associated with lower risks of all-cause, CVD, and cancer mortality (all $P < 0.001$), even after controlling for a wide range of covariates [Fig. 1 (d)-(f)]. Thus, these findings suggest that MVPA at any time of day is a protective factor against all-cause, CVD, and cancer mortality.

Fig. 1 (reduced). The associations of MVPA within time windows and fractions of MVPA within time windows with mortality risk

(d)-(f): The associations between MVPA within three time windows and mortality outcomes. The HRs were adjusted for age, sex, ethnicity, Townsend index of deprivation, recruitment center, education level, season of accelerometer wear, smoking status, alcohol intake, healthy diet score, sleep duration (< 7 hours, 7-8 hours, > 8 hours), sleep midpoint, and MVPA volume during other two time windows. **CVD:** cardiovascular disease; **MVPA:** moderate to vigorous physical activity.

We also thank the reviewer for his/her comments to remind us to provide more information about the role of MVPA volume on the associations between MVPA timing and mortality risk. Therefore, taking into account previous studies on the dosage of PA according to World Health Organization (WHO) guideline (Khurshid et al, *European Heart Journal* 2021), we chose to further investigate the potential modification effects of the WHO PA recommendation (vs. below WHO recommendation) on the associations between MVPA and mortality. As shown in **Supplementary Table 12**, there was a significant interaction effect between MVPA timing and MVPA level. The protective associations between favorable MVPA timing and all-cause and CVD mortality risks were more pronounced among individuals whose MVPA levels did not meet WHO recommendation (**Supplementary Table 17**).

Overall, MVPA is associated with a lower risk of mortality, but the timing of MVPA will moderate the association, especially for those who were less physically active (below WHO recommendation). We have made corresponding revisions throughout the manuscript and supplementary material.

Supplementary Table 12. Joint and interaction effects of timing of MVPA and MVPA level on mortality risk

Outcomes	Multiplicative interaction HR (95% CI); P	Additive interaction		
		RERI (95% CI); P	AP (95% CI); P	S (95% CI); P
All-cause mortality				
Midday-afternoon/Mixed (vs. Morning/Evening) & Below WHO recommendation ^a	1.30 (1.07-1.58); 0.009	0.40 (0.12-0.68)	0.25 (0.10-0.40)	1.66 (1.34-2.05)
CVD mortality				
Midday-afternoon/Mixed (vs. Morning/Evening) & Below WHO recommendation ^a	1.31 (0.94-1.83); 0.11	0.65 (0.22-1.07)	0.26 (0.10-0.43)	1.80 (1.07-3.03)
Cancer mortality				
Midday-afternoon/Mixed (vs. Morning/Evening) & Below WHO recommendation ^a	1.20 (0.94-1.53); 0.14	0.24 (-0.08-0.56)	0.16 (-0.03-0.36)	1.49 (1.05-2.11)

^a WHO recommendation: At least 150 minutes of moderate-intensity aerobic physical activity throughout the week, or at least 75 minutes of vigorous-intensity aerobic physical activity throughout the week, or an equivalent combination of moderate- and vigorous-intensity activity.

Multivariable Cox models were adjusted for age, sex, ethnicity, Townsend index of deprivation, recruitment center, education level, season of accelerometer wear, smoking status, alcohol intake, healthy diet score, sleep duration (< 7 hours, 7-8 hours, > 8hours), and sleep midpoint. **AP:** attributable proportion due to interaction; **CI:** confidence interval; **CVD:** cardiovascular disease; **HR:** hazard ratio; **MVPA:** moderate to vigorous physical activity; **RERI:** relative excess risk due to interaction; **S:** synergy index. According to their associations with mortality (Table 2), four timing groups were combined into two groups: 1) midday-afternoon/mixed; 2) morning/evening. This combination method facilitated the analysis and interpretation of multiplicative and additive interaction effects, which was widely used in previous studies [Shan et al. BMJ, 2018.; Huang et al. Br J Sports Med. 2021].

Supplementary Table 17. Subgroup analysis on the associations between timing of MVPA and mortality risk stratified by MVPA levels

Outcomes	PA levels	Model 1	Model 2	Model 3
		HR (95% CI); P	HR (95% CI); P	HR (95% CI); P ^b
All-cause mortality				
Midday-afternoon/Mixed (vs. Morning/Evening)	Meets WHO recommendation ^a	1.06 (0.89-1.26); 0.51	1.06 (0.89-1.26); 0.50	1.06 (0.89-1.27); 0.49
	Below WHO recommendation ^a	0.82 (0.75-0.89); <0.001	0.83 (0.76-0.91); <0.001	0.89 (0.81-0.97); 0.008 ^b
CVD mortality				
Midday-afternoon/Mixed (vs. Morning/Evening)	Meets WHO recommendation ^a	0.88 (0.65-1.19); 0.40	0.88 (0.65-1.20); 0.42	0.89 (0.66-1.21); 0.47
	Below WHO recommendation ^a	0.67 (0.58-0.77);	0.69 (0.60-0.79); <0.001	0.77 (0.66-0.88); <0.001 ^b

<0.001

Cancer mortality

Midday-afternoon/Mixed	Meets WHO recommendation ^a	1.07 (0.86-1.32); 0.55	1.07 (0.86-1.32); 0.56	1.06 (0.85-1.32); 0.59
(vs. Morning/Evening)	Below WHO recommendation ^a	0.89 (0.80-1.00); 0.06	0.90 (0.80-1.01); 0.08	0.94 (0.84-1.05); 0.29

^a WHO recommendation: At least 150 minutes of moderate-intensity aerobic physical activity throughout the week, or at least 75 minutes of vigorous-intensity aerobic physical activity throughout the week, or an equivalent combination of moderate- and vigorous-intensity activity.

^b All *P* values remained significant after multiple testing with the FDR method.

Model 1 was adjusted for age and sex. **Model 2** was adjusted as in model 1 and for ethnicity, Townsend index of deprivation, recruitment center, education level, season of accelerometer wear, smoking status, alcohol intake, and healthy diet score. **Model 3** was adjusted as in model 2 and for sleep duration (< 7 hours, 7-8 hours, > 8 hours), sleep midpoint, and total MVPA volume. **CVD**: cardiovascular disease; **HR**: hazard ratio; **MVPA**: moderate to vigorous physical activity. According to their associations with mortality (Table 2), four timing groups were combined into two groups: 1) midday-afternoon/mixed; 2) morning/evening. This combination method facilitated the analysis and interpretation of multiplicative and additive interaction effects, which was widely used in previous studies [Shan et al. *BMJ*, 2018.; Huang et al. *Br J Sports Med*. 2021]

Comment/Question 4: It is good that the authors have teased out differences between age groups and sexes. Would it be possible to do the same for BMI categories (or other measures of cardiometabolic health)? There might be differences in response which are dependent on these factors.

Response 4: According to the reviewer’s suggestions, we further ran joint associations, multiplicative and additive interaction analyses, and subgroup analyses on the cardiometabolic health indicators, including obesity (BMI ≥ 30 kg/m²) and CVDs. We found positive interaction between CVDs and unfavorable timing of MVPA (midday-afternoon/mixed vs. morning/evening) for all-cause and CVD mortality (but not cancer mortality) in both multiplicative and additive scales (**Supplementary Table 13**). However, no significant interaction between obesity and unfavorable timing of MVPA was observed (**Supplementary Table 14**). In other words, the beneficial associations of the midday-afternoon/mixed group with all-cause and CVD mortality risks were stronger among individuals with CVDs than those without. Further subgroup analyses stratified by CVDs and obesity revealed consistent results with the joint and interaction analyses (**Supplementary Table 18-19**).

We revised the manuscript as follows:

“Multiplicative and additive interaction analyses and subgroup analyses were performed on age, sex, MVPA level (meeting the WHO recommendation^{1,3} or not), CVDs, and obesity (BMI ≥ 30 kg/m²).” [**Methods (page 14; line 345-346)**]

“Synergistic effects of timing of MVPA with age, sex, MVPA level, and CVDs (but not with obesity) on

mortality risk were observed (Supplementary Table 10-14). The beneficial associations of the midday-afternoon/mixed group with all-cause and CVD mortality risks were more prominent among the elderly, less physically active (i.e., below the WHO recommendation) individuals, or those with preexisting CVDs. Additionally, the beneficial association between the midday-afternoon/mixed group and CVD mortality was enhanced among males. Subgroup analyses stratified by age, sex, MVPA level, CVDs, and obesity revealed consistent results with the interaction analyses (Fig. 2 & Supplementary Table 15-19).” [Results (page 5; line 107-113)]

“...These findings were robust to multiple testing correction and sensitivity analyses. In addition, the observed protective effects of MVPA timing were more pronounced among the elderly, males, less active individuals, or those with preexisting CVDs.

We speculate some possible reasons that may account for the beneficial effects of MVPA timing. One possible reason could be related to circadian modulation of cardiometabolic reactivity to physical exercise. For instance, the unfavorable timing of MVPA we found is very close to the morning (~6:00-12:00) and evening (~18:00-22:00) peaks for CVD risk^{27,28}. In line with this, human trials found time-of-day specific cardiovascular reactivity to physical exercises, such as the greatest vagal withdrawal at ~9:00²⁹, peaks of catecholamine reactivity at ~9:00 and ~21:00²⁹, and faster recovery of systolic blood pressure after exercise in the late afternoon (~17:00) than in the early morning (~8:30)³⁰. In addition, this explanation may also be supported by our findings of stronger associations between unfavorable timing of MVPA and higher mortality risk among individuals with preexisting CVDs. ...” [Discussion (page 5-6; line 120-145)]

“Another interesting finding of this present study is that age, sex, MVPA levels, and preexisting CVDs may modify the associations between MVPA timing and mortality risk, particularly CVD mortality. A similar synergistic effect between sex and MVPA timing has been reported in a previous cross-sectional study of patients with type 2 diabetes in which the association between morning MVPA and a higher coronary heart disease risk score was only observed in men²⁰.

The possible reasons might be related to sex-specific and age-specific physiological responses to PA^{32,33}. There may be a concern that potential variations in total MVPA across age and sex subgroups may contribute to the observed interaction effects, but further adjustment for total MVPA did not change the results, suggesting that total MVPA is unlikely to contribute to age and sex effects. In addition, the finding of synergistic effects of

MVPA timing with MVPA levels and preexisting CVDs also extends our knowledge in this field. Further studies are needed to explore the mechanisms underlying these synergistic effects.

Our findings have several implications. ...Third, this study may help to identify individuals (i.e., the elderly, males, less active individuals, and those with preexisting CVDs) who could have greater beneficial effects of choosing a more favorable MVPA timing. This may also provide insights for planning more effective intervention strategies.

MVPA is associated with lower risks of all-cause, CVD, and cancer mortality regardless of the time of day. Furthermore, midday-afternoon and mixed MVPA timing had greater beneficial effects against all-cause and CVD mortality, particularly among the older, male, less physically active individuals, or those with preexisting CVDs. Future experimental studies are needed to validate the potential effects of MVPA timing on health and longevity.” [Discussion (page 7-9; line 154-203)]

Supplementary Table 13. Joint and interaction effects of timing of MVPA and CVDs on mortality risk

Outcomes	Multiplicative interaction HR (95% CI); P	Additive interaction		
		RERI (95% CI); P	AP (95% CI); P	S (95% CI); P
All-cause mortality				
Midday-afternoon/Mixed (vs. Morning/Evening) & CVDs	1.17 (1.00-1.37); <0.05	0.29 (0.07-0.52)	0.16 (0.04-0.27)	1.51 (1.07-2.12)
CVD mortality				
Midday-afternoon/Mixed (vs. Morning/Evening) & CVDs	1.32 (1.01-1.71); 0.04	0.94 (0.42-1.47)	0.28 (0.15-0.42)	1.68 (1.24-2.28)
Cancer mortality				
Midday-afternoon/Mixed (vs. Morning/Evening) & CVDs	1.09 (0.88-1.34); 0.43	0.12 (-0.14-0.38)	0.08 (-0.09-0.25)	1.36 (0.69-2.69)

Multivariable Cox models were adjusted for age, sex, ethnicity, Townsend index of deprivation, recruitment center, education level, season of accelerometer wear, smoking status, alcohol intake, healthy diet score, sleep duration (< 7 hours, 7-8 hours, > 8hours), sleep midpoint, and total MVPA volume. **AP**: attributable proportion due to interaction; **CI**: confidence interval; **CVDs**: cardiovascular diseases; **HR**: hazard ratio; **MVPA**: moderate to vigorous physical activity; **RERI**: relative excess risk due to interaction; **S**: synergy index. According to their associations with mortality (Table 2), four timing groups were combined into two groups: 1) midday-afternoon/mixed; 2) morning/evening. This combination method facilitated the analysis and interpretation of multiplicative and additive interaction effects, which was widely used in previous studies [Shan et al. BMJ, 2018.; Huang et al. Br J Sports Med. 2021].

Supplementary Table 14. Joint and interaction effects of timing of MVPA and obesity on mortality risk

Outcomes	Multiplicative interaction	Additive interaction		
	HR (95% CI); P	RERI (95% CI); P	AP (95% CI); P	S (95% CI); P
All-cause mortality				
Midday-afternoon/Mixed (vs. Morning/Evening) & Obesity	1.07 (0.90-1.27); 0.43	0.13 (-0.10-0.35)	0.08 (-0.06-0.23)	1.33 (0.80-2.22)
CVD mortality				
Midday-afternoon/Mixed (vs. Morning/Evening) & Obesity	1.11 (0.85-1.44); 0.45	0.34 (-0.09-0.78)	0.16 (-0.03-0.35)	1.44 (0.89-2.33)
Cancer mortality				
Midday-afternoon/Mixed (vs. Morning/Evening) & Obesity	1.09 (0.87-1.36); 0.45	0.12 (-0.16-0.40)	0.09(-0.10-0.28)	1.45 (0.60-3.50)

Multivariable Cox models were adjusted for age, sex, ethnicity, Townsend index of deprivation, recruitment center, education level, season of accelerometer wear, smoking status, alcohol intake, healthy diet score, sleep duration (< 7 hours, 7-8 hours, > 8hours), sleep midpoint, and total MVPA volume. **AP**: attributable proportion due to interaction; **CI**: confidence interval; **CVD**: cardiovascular disease; **HR**: hazard ratio; **MVPA**: moderate to vigorous physical activity; **RERI**: relative excess risk due to interaction; **S**: synergy index. According to their associations with mortality (Table 2), four timing groups were combined into two groups: 1) midday-afternoon/mixed; 2) morning/evening. This combination method facilitated the analysis and interpretation of multiplicative and additive interaction effects, which was widely used in previous studies [Shan et al. BMJ, 2018.; Huang et al. Br J Sports Med. 2021].

Supplementary Table 18 (reduced). Subgroup analysis on the associations between timing of MVPA and mortality risk stratified by CVDs

Outcomes	CVD categories	Model 1	Model 2	Model 3
		HR (95% CI); P	HR (95% CI); P	HR (95% CI); P ^a
All-cause mortality				
Midday-afternoon/Mixed (vs. Morning/Evening)	No CVDs ^a	0.93 (0.84-1.04); 0.20	0.94 (0.84-1.04); 0.24	0.96 (0.86-1.07); 0.42
	CVDs ^a	0.78 (0.70-0.87); <0.001	0.80 (0.71-0.89); <0.001	0.84 (0.75-0.94); 0.002 ^a
CVD mortality				
Midday-afternoon/Mixed (vs. Morning/Evening)	No CVDs ^a	0.84 (0.68-1.03); 0.09	0.85 (0.69-1.04); 0.11	0.89 (0.72-1.09); 0.26
	CVDs ^a	0.62 (0.53-0.73); <0.001	0.64 (0.55-0.76); <0.001	0.69 (0.58-0.81); <0.001 ^a
Cancer mortality				
Midday-afternoon/Mixed (vs. Morning/Evening)	No CVDs ^a	0.96 (0.84-1.10); 0.56	0.97 (0.85-1.10); 0.62	0.98 (0.85-1.12); 0.72
	CVDs ^a	0.87 (0.75-1.02); 0.09	0.89 (0.76-1.04); 0.15	0.92 (0.78-1.07); 0.27

^a All P values remained significant after multiple testing with the FDR method.

Model 1 was adjusted for age and sex. **Model 2** was adjusted as in model 1 and for ethnicity, Townsend index of deprivation, recruitment center, education level, season of accelerometer wear, smoking status, alcohol intake, and healthy diet score. **Model 3** was adjusted as in model 2 and for sleep duration (< 7 hours, 7-8 hours, > 8 hours), sleep midpoint, and total MVPA volume. **CVDs**: cardiovascular diseases; **HR**: hazard ratio; **MVPA**: moderate to vigorous physical activity. According to their associations with mortality (Table 2), four timing groups were combined into two groups: 1) midday-afternoon/mixed; 2) morning/evening. This combination method facilitated the analysis and interpretation of multiplicative and additive interaction effects, which was widely used in previous studies [Shan et al. BMJ, 2018.; Huang et al. Br J Sports Med. 2021]

Supplementary Table 19 (reduced). Subgroup analysis on the associations between timing of MVPA and mortality risk stratified by obesity

Outcomes	Obesity categories	Model 1 HR (95% CI); P	Model 2 HR (95% CI); P	Model 3 HR (95% CI); P ^a
All-cause mortality				
Midday-afternoon/Mixed (vs. Morning/Evening)	No obesity ^a	0.88 (0.80-0.97); 0.008	0.89 (0.81-0.98); 0.02	0.92 (0.84-1.01); 0.08
	Obesity ^a	0.84 (0.73-0.97); 0.01	0.85 (0.74-0.98); 0.02	0.87 (0.75-1.00); <0.05
CVD mortality				
Midday-afternoon/Mixed (vs. Morning/Evening)	No obesity ^a	0.73 (0.63-0.86); <0.001	0.75 (0.64-0.88); <0.001	0.76 (0.68-0.93); 0.005 ^a
	Obesity ^a	0.68 (0.55-0.83); <0.001	0.68 (0.56-0.84); <0.001	0.72 (0.59-0.90); 0.003 ^a
Cancer mortality				
Midday-afternoon/Mixed (vs. Morning/Evening)	No obesity ^a	0.95 (0.85-1.08); 0.44	0.96 (0.85-1.09); 0.54	0.97 (0.86-1.10); 0.67
	Obesity ^a	0.89 (0.74-1.08); 0.23	0.90 (0.74-1.08); 0.25	0.92 (0.76-1.11); 0.38

^a All P values remained significant after multiple testing with the FDR method.

Model 1 was adjusted for age and sex. **Model 2** was adjusted as in model 1 and for ethnicity, Townsend index of deprivation, recruitment center, education level, season of accelerometer wear, smoking status, alcohol intake, and healthy diet score. **Model 3** was adjusted as in model 2 and for sleep duration (< 7 hours, 7-8 hours, > 8 hours), sleep midpoint, and total MVPA volume. **CVD**: cardiovascular disease; **HR**: hazard ratio; **MVPA**: moderate to vigorous physical activity. According to their associations with mortality (Table 2), four timing groups were combined into two groups: 1) midday-afternoon/mixed; 2) morning/evening. This combination method facilitated the analysis and interpretation of multiplicative and additive interaction effects, which was widely used in previous studies [Shan et al. *BMJ*, 2018.; Huang et al. *Br J Sports Med*. 2021]

Comment/Question 5: Page 10, lines 236-237: I suggest modifying your statement about the implications of the findings: I will say that these data suggest that timing of MVPA may have impact on health (based on the fact that these are observational data and there might be other confounding variables that you did not adjust for). There is clearly some confounding present as the estimates are less clear the more you adjust in your models. Do you for example have employment status of your participants?

Response 5: We thank the reviewer for pointing out this issue. Although we have controlled for a wide range of confounders in the analyses, there were still possible unmeasured or residual confounding factors. Employment status was measured at the initial visit of the UK Biobank, 5.6 years before the present study baseline. More importantly, a previous study demonstrated that employment status is unstable over years among UK Biobank participants with accelerometer data (mean age: 62 years) (Strain et al. *Nature Medicine*. 2020).

Therefore, we did not include employment status as a covariate in Cox regression models in this study.

As suggested, we have revised the implications of our findings as follows: “*Our findings have several implications. First, the results of this study suggest that the timing of MVPA may have an impact on health.*” In addition, we added this as a limitation of our study: “*Fourth, despite the inclusion of a wide range of confounders in the analyses, residual or unmeasured confounding cannot be ruled out in our study, such as employment status, mode of PA, and meal timing. The observational nature of this study precludes definitive causal relationships.*” [Discussion (page 7-8; line 166-167 & 193-196)]

MINOR COMMENTS:

Comment/Question 6: I would rephrase the second sentence of the introduction (replacing suggested with suggest).

Response 6: As suggested, “Compelling evidence suggested ...” has been changed to “Compelling evidence suggest”.

Comment/Question 7: In the results, I would move the total number of deaths etc (now on page 7, second paragraph) up, as this is important information to have in mind when reading all the results.

Response 7: According to the reviewer’s suggestions, we moved the sentence “*Over a median of 7.0 years (638,825 person-years), 3088 (3.35%) participants died from all-cause, 1076 (1.17%) from CVD, and 1872 (2.03%) from cancer.*” to the first paragraph of Results. [page 3; line 49-51]

Comment/Question 8: In the discussion, I do not see the relevance of bringing in the findings from reference 31.

Response 8: We thank the reviewer for pointing out this issue. We have revised this part as follows:

“We speculate some possible reasons that may account for the beneficial effects of MVPA timing. ... Another possibility is that the potential differences in exercise mode across timing groups may contribute to the associations between MVPA timing and health risk. It has been reported that combined training can cause a greater decrease in body fat than aerobic or resistance training alone³¹. However, time-of-day specific data of both PA mode and behavioral/environmental factors were not available in this study, and we were not able to

testify these hypotheses.” [Discussion (page 7; line 148-152)]

Comment/Question 9: It is unclear to me what 'it' refers to in line 242 on page 10.

Response 9: We revised the sentence “*Second, the timing of MVPA can be easily measured objectively and is highly reproducible due to the development of wearable devices and technology⁶, thus, it has the potential to improve public health.*” to “*Second, the timing of MVPA can be easily measured objectively and is highly reproducible due to the development of wearable devices and technology⁶. The accessibility and convenience of accelerometer measurement make the impact of MVPA timing and measurement of MVPA more meaningful and actionable in public health.*” [Discussion (page 7; line 170-173)]

****Responses to the Reviewer #2****

Comment/Question 1: General Comments: This is a well documented secondary analysis of the UK Biobank database seeking to examine the association of moderate to vigorous physical activity (MVPA) with total, cardiovascular and selected cancer mortality. The uniqueness of this study is the examination of objectively measured MVPA (wearables/accelerometry) among 92K+ adults and specifically whether a daily time period (morning, mid-day, and evening) vary among these associations. The methods are well described, use of an excellent and unique data source, and clear study rationale supported by a thorough literature review. The analytic approach is most appropriate in using Cox proportional hazards modeling and expressing the data using person years with calculated hazard ratios and 95% CIs. The results are well presented complemented by both narrative and tabular data. Supplementary figures and tables provide further insights into the results with a review of the sensitivity analyses. The discussion is concise and stays within the bounds of the data presented. Further explanation in the limitations section may shed further light on the MVPA volume differences that exist between the morning (referent) and mid-day and evening groups. Where the morning group presents with a visibly lower volume of MVPA than the mid-day and mixed groups, with the evening group showing the lowest volume.

Response 1: We thank the reviewer for his/her assessment and positive comments.

Comment/Question 2: Did the authors seek to stratify by volume of MVPA across each of the time groups

-- hence examining each time period controlling for volume, or did I miss this adjustment.

Response 2: In the previous version, we have controlled for a wide range of potential confounding factors including total volume of MVPA. We found that the significant associations between MVPA timing and mortality risk were independent of total volume of MVPA (main analysis-**Table 2**; Sensitivity analyses-**Table 3** and **Supplementary Table 1-9**).

We appreciate the reviewer for reminding us to testify whether MVPA level modifies the associations between MVPA timing and mortality risk. Therefore, taking into account previous studies on the dosage of PA according to World Health Organization (WHO) guideline (Khurshid et al, *European Heart Journal* 2021), we chose to further investigate the potential modification effects of the WHO PA recommendation (vs. below WHO recommendation) on the associations between MVPA and mortality. As shown in **Supplementary Table 12**, there was a significant interaction effect between MVPA timing and MVPA level. The protective associations between favorable MVPA timing and all-cause and CVD mortality risks were more pronounced among individuals whose MVPA levels did not meet WHO recommendation (**Supplementary Table 17**). Overall, MVPA is associated with a lower risk of mortality, but the timing of MVPA will moderate the association, especially for those who were less physically active (below WHO recommendation). We have made corresponding revisions throughout the manuscript and supplementary material. For more details, please also refer to our responses to Comment 3 of Reviewer 1.

Supplementary Table 12. Joint and interaction effects of timing of MVPA and MVPA level on mortality risk

Outcomes	Multiplicative interaction	Additive interaction		
	HR (95% CI); P	RERI (95% CI); P	AP (95% CI); P	S (95% CI); P
All-cause mortality				
Midday-afternoon/Mixed (vs. Morning/Evening) & Below WHO recommendation ^a	1.30 (1.07-1.58); 0.009	0.40 (0.12-0.68)	0.25 (0.10-0.40)	1.66 (1.34-2.05)
CVD mortality				
Midday-afternoon/Mixed (vs. Morning/Evening) & Below WHO recommendation ^a	1.31 (0.94-1.83); 0.11	0.65 (0.22-1.07)	0.26 (0.10-0.43)	1.80 (1.07-3.03)
Cancer mortality				
Midday-afternoon/Mixed (vs. Morning/Evening) & Below WHO recommendation ^a	1.20 (0.94-1.53); 0.14	0.24 (-0.08-0.56)	0.16 (-0.03-0.36)	1.49 (1.05-2.11)

^a WHO recommendation: At least 150 minutes of moderate-intensity aerobic physical activity throughout the week, or at least 75 minutes of vigorous-intensity aerobic physical activity throughout the week, or an equivalent combination of moderate- and vigorous-intensity activity.

Multivariable Cox models were adjusted for age, sex, ethnicity, Townsend index of deprivation, recruitment center, education level, season of accelerometer wear, smoking status, alcohol intake, healthy diet score, sleep duration (< 7 hours, 7-8 hours, > 8hours), and sleep midpoint. **AP**: attributable proportion due to interaction; **CI**: confidence interval; **CVD**: cardiovascular disease; **HR**: hazard ratio; **MVPA**: moderate to vigorous physical activity; **RERI**: relative excess risk due to interaction; **S**: synergy index. According to their associations with mortality (Table 2), four timing groups were combined into two groups: 1) midday-afternoon/mixed; 2) morning/evening. This combination method facilitated the analysis and interpretation of multiplicative and additive interaction effects, which was widely used in previous studies [Shan et al. BMJ, 2018.; Huang et al. Br J Sports Med. 2021].

Supplementary Table 17. Subgroup analysis on the associations between timing of MVPA and mortality risk stratified by MVPA levels

Outcomes	PA levels	Model 1	Model 2	Model 3
		HR (95% CI); P	HR (95% CI); P	HR (95% CI); P ^b
All-cause mortality				
Midday-afternoon/Mixed	Meets WHO recommendation ^a	1.06 (0.89-1.26); 0.51	1.06 (0.89-1.26); 0.50	1.06 (0.89-1.27); 0.49
(vs. Morning/Evening)	Below WHO recommendation ^a	0.82 (0.75-0.89); <0.001	0.83 (0.76-0.91); <0.001	0.89 (0.81-0.97); 0.008 ^b
CVD mortality				
Midday-afternoon/Mixed	Meets WHO recommendation ^a	0.88 (0.65-1.19); 0.40	0.88 (0.65-1.20); 0.42	0.89 (0.66-1.21); 0.47
(vs. Morning/Evening)	Below WHO recommendation ^a	0.67 (0.58-0.77); <0.001	0.69 (0.60-0.79); <0.001	0.77 (0.66-0.88); <0.001 ^b
Cancer mortality				
Midday-afternoon/Mixed	Meets WHO recommendation ^a	1.07 (0.86-1.32); 0.55	1.07 (0.86-1.32); 0.56	1.06 (0.85-1.32); 0.59
(vs. Morning/Evening)	Below WHO recommendation ^a	0.89 (0.80-1.00); 0.06	0.90 (0.80-1.01); 0.08	0.94 (0.84-1.05); 0.29

^a WHO recommendation: At least 150 minutes of moderate-intensity aerobic physical activity throughout the week, or at least 75 minutes of vigorous-intensity aerobic physical activity throughout the week, or an equivalent combination of moderate- and vigorous-intensity activity.

^b All P values remained significant after multiple testing with the FDR method.

Model 1 was adjusted for age and sex. **Model 2** was adjusted as in model 1 and for ethnicity, Townsend index of deprivation, recruitment center, education level, season of accelerometer wear, smoking status, alcohol intake, and healthy diet score. **Model 3** was adjusted as in model 2 and for sleep duration (< 7 hours, 7-8 hours, > 8 hours), sleep midpoint, and total MVPA volume. **CVD**: cardiovascular disease; **HR**: hazard ratio; **MVPA**: moderate to vigorous physical activity. According to their associations with mortality (Table 2), four timing groups were combined into two groups: 1) midday-afternoon/mixed; 2) morning/evening. This combination method facilitated the analysis and interpretation of multiplicative and additive interaction effects, which was widely used in previous studies [Shan et al. BMJ, 2018.; Huang et al. Br J Sports Med. 2021]

Comment/Question 3: The most significant and satisfying finding is that regardless of time of day, those participating in significant doses of MVPA accrued a protective effect from pre-mature mortality.

Response 3: We agree with the reviewer that the findings of the beneficial effects of MVPA regardless of time of day are interesting and significant to public health.

****Responses to the Reviewer #3****

Comment/Question 1: This was a generally well-written report on an interesting and important investigation using UK Biobank data. The authors performed a thorough analysis of the relationship between objectively measured MVPA timing and all-cause, CVD, and cancer mortality. This addresses a substantial gap in current knowledge. The manuscript was enhanced by thoughtful sensitivity and supplemental analyses. I do, however, have some comments and questions for the authors, as listed below.

Response 1: We appreciate the reviewer's assessment and positive comments.

Comment/Question 2: -Lines 73-75 – Supplementary Fig. 5 is referred to in support of the time windows. Were subjects required to accumulate 50+% of their MVPA in a 2-hour period in order to be classified in one of the non-mixed categories? This post-hoc approach would seem to make it more likely to generate groups with significant differences.

Response 2: Exposure-dependent methods, such as equally spaced intervals, are generally arbitrary and may not help assess a variable's actual predictive value. In contrast, the outcome-based methods allow an "optimal" cutoff to be estimated (Williams et al. *Technical Report Series of Mayo Clinic* 2006; Contal et al. *Computational Statistics & Data Analysis* 1999; Altman et al. *J Natl Cancer Inst* 1994). For outcome-based methods, researchers generally used linear/non-linear curves to illustrate the associations between continuous variables (exposures) and outcomes. Then, the continuous variables were always converted to categorical variables using the cutoffs based on quartiles/quintiles (for linear curves) or change points (for non-linear curves) (Ho et al. *BMJ* 2020). In addition, when splitting the continuous variables using outcome-based methods, researchers may commonly consider other factors, such as the sample size of each category, accuracy, simplification, and comparisons with other studies.

Therefore, in this study, we chose to run our exploratory analysis to identify time window boundaries using the 'outcome-based' method. In addition, to balance sample size and accuracy (too small sample size using 1-h periods and too low 'resolution' using 3-h periods), we chose the 2-h time window intervals (but 3-h period only for 21:00-24:00 due to sampling size issue) in the exploratory analyses. The 50% method of assigning timing groups was similar to that previously used (Qian et al., *Diabetes Care* 2021), and this method prevents the participants from being assigned to multiple timing groups. If over 50% of total daily MVPA occurred

during the same 2-h period, participants were assigned to the corresponding groups. For those who spent less than 50% of the total daily MVPA in any of the 2-h time windows, we assigned them to the mixed group. As shown in **Supplementary Fig. 5**, which mimics the non-linear exposure-outcome curves variable we mentioned above, two change points at 11:00 and 17:00 were consistently observed for mortality outcomes. Thus, morning (05:00-11:00), midday to afternoon (11:00-17:00), and evening (17:00-24:00) time windows were used in subsequent MVPA timing grouping and statistical analyses.

We revised the manuscript to provide more details about this analysis as follows:

*“Then, we ran the exploratory analyses to determine the appropriate boundaries for the time windows used for the main analyses. Exposure-dependent methods, such as the equally spaced intervals (which does not appear to be clinically driven), are generally arbitrary and may not be helpful in assessing a variable’s actual predictive value²⁶. In contrast, the outcome-based methods allow an “optimal” cutoff to be estimated^{41,42}. Therefore, this study ran an exploratory analysis for the identification of time window boundaries using the ‘outcome-based’ method. In addition, to balance sample size and accuracy, we chose the 2-h time window intervals (3-h only for the 21:00-24:00 period) in the exploratory analyses. The 50% method of assigning timing groups was similar to that previously used²⁰, and this method avoids the participants being assigned to multiple timing groups. If over 50% of total daily MVPA occurred during the same 2-h period, participants were assigned to the corresponding groups. For those who spent less than 50% of total daily MVPA in any of the 2-h time windows, we assigned them to the mixed group. As shown in Supplementary Fig. 5, which mimics the non-linear exposure-outcome curves, two change points at 11:00 and 17:00 were consistently observed for all mortality outcomes. Compared with the mixed group, the 2-h timing groups of the morning (05:00-11:00) and evening (17:00-24:00) periods seemed to have higher mortality risks. The 2-h timing groups of the midday to afternoon period (11:00-17:00) presented comparable mortality risks with the mixed group (Supplementary Fig. 5). Finally, morning (05:00-11:00), midday to afternoon (11:00-17:00), and evening (17:00-24:00) time windows were used in subsequent MVPA timing grouping and statistical analyses.”***[Methods (page 11; line 263-279)]**

Supplementary Fig. 5. The exploratory analysis on the associations between timing of MVPA and mortality risk

^a All *P* values remained significant after multiple testing with FDR method. **CVDs**: cardiovascular diseases. HRs were adjusted for age, sex, ethnicity, Townsend index of deprivation, recruitment center, education level, season of accelerometer wear, healthy diet score, smoking status, alcohol intake, sleep duration (< 7 hours, 7-8 hours, > 8hours), sleep midpoint, and total MVPA volume (**Model 3**). **CVD**: cardiovascular disease; **HR**: hazard ratio; **MVPA**: moderate to vigorous physical activity.

Comment/Question 3: -Line 89 – Analyses were adjusted for season of accelerometer wear. However, this does not seem sufficient to account for potential misclassification of PA timing due to season of measurement.

Response 3: We thank the reviewer for pointing out this issue. The times of the year for accelerometer measurement may cause misclassification of MVPA timing in two major ways: daylight saving time and the long-term changes in physical activity over seasons/months. In the previous manuscript, we performed sensitivity analysis by excluding participants who wore accelerometers during the daylight saving time transition (**Supplementary Table 4**) and found that the associations between MVPA timing and mortality risks were generally consistent with the main analysis (**Table 2**). To further test the robustness of our findings, we conducted sensitivity analysis by controlling for month of accelerometer wear. As shown in **Supplementary Table 5**, there were no substantial changes in our main findings. Nevertheless, these sensitivity analyses may not be sufficient to account for potential misclassification of PA timing due to the times of the year for accelerometer measurement. Therefore, we revised our manuscript as follows:

“However, this study has some limitations. ... Third, although 7-day monitoring periods have been routinely used in PA monitor studies because they provide the opportunity to sample PA on both week and weekend days and achieve intra-class correlations of greater than 80% in most populations³⁵, it is still unclear whether a 7-day accelerometer measurement is representative of longer-term behaviors³⁶, especially for PA timing. In addition, although our findings were robust to daylight saving time and adjustment of season (even month) of accelerometer wear, these sensitivity analyses may not be sufficient to account for potential misclassification or seasonal variation of PA timing. These situations highlight the importance of repeated and long-term administration of accelerometers in future studies.”[Discussion (page 8; line 186-193)]

Supplementary Table 4. Sensitivity analysis on the associations between timing of MVPA and mortality risk by excluding participants who wore accelerometers during the daylight saving time transition

Outcomes	Events	n	Person-years	Model 1 HR (95% CI); P	Model 2 HR (95% CI); P	Model 3 HR (95% CI); P ^a
All-cause mortality	1849	88,027	450,358			
Morning	394	15,132	77,198	1.00 (reference)	1.00 (reference)	1.00 (reference)
Midday-afternoon	844	39,213	200,043	0.87 (0.79-0.95); 0.004	0.89 (0.81-0.98); 0.01	0.90 (0.82-0.99); 0.04
Evening	154	7955	40,806	1.08 (0.93-1.25); 0.34	1.08 (0.93-1.26); 0.29	1.01 (0.86-1.17); 0.94
Mixed	457	25,727	132,311	0.88 (0.79-0.98); 0.02	0.89 (0.80-0.99); 0.04	0.89 (0.80-1.00); 0.04
CVD mortality	624	88,027	450,358			

Morning	158	15,132	77,198	1.00 (reference)	1.00 (reference)	1.00 (reference)
Midday-afternoon	270	39,213	200,043	0.67 (0.57-0.78); <0.001	0.69 (0.60-0.81); <0.001	0.73 (0.63-0.85); <0.001 ^a
Evening	55	7955	40,806	0.97 (0.76-1.24); 0.82	0.99 (0.77-1.26); 0.93	0.90 (0.71-1.15); 0.42
Mixed	141	25,727	132,311	0.72 (0.60-0.86); <0.001	0.73 (0.61-0.88); <0.001	0.74 (0.62-0.89); 0.001 ^a
Cancer mortality	1176	88,027	450358			
Morning	225	15,132	77,198	1.00 (reference)	1.00 (reference)	1.00 (reference)
Midday-afternoon	561	39,213	200,043	0.97 (0.85-1.10); 0.62	0.98 (0.87-1.12); 0.81	0.99 (0.87-1.12); 0.86
Evening	100	7955	40,806	1.16 (0.96-1.41); 0.13	1.17 (0.96-1.42); 0.12	1.10 (0.90-1.33); 0.35
Mixed	290	25 727	132 311	0.96 (0.83-1.11); 0.56	0.97 (0.84-1.12); 0.66	0.97 (0.84-1.12); 0.65

^a All *P* values remained significant after multiple testing with the FDR method. **Model 1** was adjusted for age and sex. **Model 2** was adjusted as in model 1 and for ethnicity, Townsend index of deprivation, recruitment center, education level, season of accelerometer wear, smoking status, and alcohol intake, healthy diet score. **Model 3** was adjusted as in model 2 and for sleep duration (< 7 hours, 7-8 hours, > 8 hours), sleep midpoint, and total MVPA volume. **CVD**: cardiovascular disease; **HR**: hazard ratio; **MVPA**: moderate to vigorous physical activity.

Supplementary Table 5. Sensitivity analysis on the associations between timing of MVPA and mortality risk by controlling for month of accelerometer wear

Outcomes	Events	n	Person-years	Model 1 HR (95% CI); P	Model 2 HR (95% CI); P	Model 3 HR (95% CI); P ^a
All-cause mortality	3088	92,139	638,825			
Morning	652	15,865	109,537	1.00 (reference)	1.00 (reference)	1.00 (reference)
Midday-afternoon	1432	41,125	284,481	0.85 (0.78-0.94); <0.001	0.87 (0.79-0.95); 0.003	0.89 (0.81-0.97); 0.01 ^a
Evening	249	8307	57,768	1.05 (0.91-1.21); 0.54	1.05 (0.90-1.21); 0.53	0.97 (0.84-1.13); 0.71
Mixed	755	26,842	187,039	0.88 (0.79-0.97); 0.01	0.89 (0.80-0.98); 0.02	0.89 (0.80-0.99); 0.03 ^a
CVD mortality	1076	92,139	638,825			
Morning	274	15,865	109,537	1.00 (reference)	1.00 (reference)	1.00 (reference)
Midday-afternoon	464	41,125	284,481	0.66 (0.57-0.77); <0.001	0.68 (0.59-0.80); <0.001	0.72 (0.62-0.84); <0.001 ^a
Evening	90	8307	57,768	0.94 (0.74-1.20); 0.64	0.96 (0.76-1.22); 0.75	0.87 (0.68-1.11); 0.27
Mixed	248	26,842	187,039	0.72 (0.60-0.85); <0.001	0.74 (0.62-0.87); <0.001	0.74 (0.62-0.88); <0.001 ^a
Cancer mortality	1872	92,139	638,825			
Morning	362	15,865	109,537	1.00 (reference)	1.00 (reference)	1.00 (reference)
Midday-afternoon	888	41,125	284,481	0.95 (0.84-1.08); 0.42	0.96 (0.85-1.09); 0.57	0.97 (0.86-1.10); 0.62
Evening	154	8307	57,768	1.14 (0.94-1.37); 0.19	1.13 (0.93-1.37); 0.21	1.06 (0.88-1.29); 0.53
Mixed	468	26,842	187,039	0.96 (0.83-1.10); 0.53	0.96 (0.84-1.10); 0.59	0.96 (0.84-1.10); 0.57

^a All *P* values remained significant after multiple testing with the FDR method.

Model 1 was adjusted for age and sex. Model 2 was adjusted as in model 1 and for ethnicity, Townsend index of deprivation, recruitment center, education level, **month of accelerometer wear**, healthy diet score, smoking status, and alcohol intake. Model 3 was adjusted as in model 2 and for sleep duration (< 7 hours, 7-8 hours, > 8 hours), sleep midpoint, and total MVPA volumes. **CVD**: cardiovascular disease; **HR**: hazard ratio; **MVPA**: moderate to vigorous physical activity.

Comment/Question 4: -Lines 91-92. Based on Supplementary Figure 3, approximately 30,000 participants had a second study interview – which may have occurred before or after the accelerometry

measurement period. Can the authors clarify what information was collected at these subsequent interviews and how they were used in the analysis. For example, did some subjects have time-varying covariates entered in the models?

Response 4: We thank the reviewer for his/her comments to remind us to clearly describe the repeated measurements of some covariates, including education level, smoking status, alcohol consumption, healthy diet score, obesity, diabetes history, longstanding illness, and cancer history. The values of these covariates were obtained from touchscreen questionnaires at the time-point closest to the accelerometry (**Supplementary fig 3**). Therefore, we did not have time-varying covariates in the Cox regression models. It should be noted that most of these covariates, such as education, smoking status, alcohol consumption, diet, and BMI, were stable over years, as suggested by a previous UK Biobank study (Strain et al, *Nature Medicine* 2020). Therefore, this time lag is unlikely to weaken the validity of our findings.

We revised the manuscript as follows:

(1) *“The values of some covariates, including education level, smoking status, alcohol consumption, healthy diet score, obesity, diabetes history, longstanding illness, and cancer history, were obtained from touchscreen questionnaires at the time-point closest to the accelerometry (Supplementary Fig. 6).”* [Methods (page 13; line 311-314)]

(2) We revised the figure legend of *Supplementary Fig. 6*.

Supplementary fig 6. Timeline of some covariates collection

The covariates with repeated measurements include education level, smoking status, alcohol consumption, healthy diet score, obesity, diabetes history, longstanding illness, and cancer history were obtained from touchscreen questionnaires at the time-point closest to the accelerometry.

Comment/Question 5: -Lines 107-108. Did assessment of health-related variables occur only at the recruitment interview or at other times. If done at the recruitment interview, might these pre-existing conditions influence the timing and amount of MVPA during the later accelerometer wear period? If so, how would that influence the interpretation of the models which include these health-related variables?

Response 5: The information of health-related variables was obtained at the recruitment interview. According to the reviewer's suggestions, we further ran multiplicative and additive interaction analyses and subgroup analyses on the cardiometabolic health indicators (including obesity and CVD, as suggested by Reviewer 1). We found positive interaction between CVDs and unfavorable timing of MVPA (midday-afternoon/mixed vs. morning/evening) for all-cause and CVD mortality (but not cancer mortality) in both multiplicative and additive scales (**Supplementary Table 13**). Nevertheless, no significant interaction between obesity and unfavorable timing of MVPA was observed (**Supplementary Table 14**). In other words, the beneficial associations of the midday-afternoon/mixed group with all-cause and CVD mortality risks were stronger among individuals with CVDs than those without. Subgroup analyses stratified by CVDs and obesity revealed consistent results with the joint and interaction analyses (**Supplementary Table 18-19**). We have made corresponding revisions throughout the manuscript and supplementary material. For the details, please also refer to our responses to Comment 4 of Reviewer 1.

Comment/Question 6: -Lines 112-113. In the main analysis, was there a minimum number of accelerometer wear days required for inclusion? How was a valid accelerometer wear day defined?

Response 6: In the main analysis, we included 92,139 participants with at least 3 days of accelerometry data. This inclusion criterion (Field ID: 90015 = 1) was defined by the UK Biobank accelerometer expert working group after finding that 3 days of wear were needed to be within 10% of a complete seven-day measure in missing data simulations on 29,765 participants who had perfect wear time compliance (Doherty et al, *PLoS One* 2017).

Comment/Question 7: -Line 121. I think you mean “process” instead of “progress”.

Response 7: We apologize for this typo. Following the format requirements of *Nature Communications*, we deleted the whole “Patient and public involvement” part, and this typo has been removed.

Comment/Question 8: -Figure 1. In the legend, both the morning and evening times appear to be

represented by solid lines (darker for morning). However, there is only one solid line in the graphs. Please adjust the legend to clarify.

Response 8: We thank the reviewer for reminding us. In the revised manuscript, we have updated the Fig. 1 to clearly show the lines.

Fig. 1. The associations of total MVPA volume, MVPA within time windows, and fractions of MVPA within time windows with mortality risk

(a)-(c): The associations between total MVPA volume and mortality outcomes. The HRs were adjusted for age, sex, ethnicity, Townsend index of deprivation, recruitment center, education level, season of accelerometer wear, smoking status, alcohol intake, healthy diet score, sleep duration (< 7 hours, 7-8 hours, > 8hours), and sleep midpoint. **(d)-(f):** The associations between MVPA

within three time windows and mortality outcomes. The HRs were adjusted for age, sex, ethnicity, Townsend index of deprivation, recruitment center, education level, season of accelerometer wear, smoking status, alcohol intake, healthy diet score, sleep duration (< 7 hours, 7-8 hours, > 8hours), sleep midpoint, and MVPA volume during other two time windows. **(g)-(i)**: The associations between the fractions of MVPA within three time windows with mortality risk (indicating MVPA timing effects). The HRs were adjusted for age, sex, ethnicity, Townsend index of deprivation, recruitment center, education level, season of accelerometer wear, healthy diet score, smoking status, alcohol intake, sleep duration (< 7 hours, 7-8 hours, > 8hours), sleep midpoint, and total MVPA volume. **CVD**: cardiovascular disease; **MVPA**: moderate to vigorous physical activity.

Comment/Question 9: -Figure 1. Please confirm that graphs G-I were additionally adjusted for only total MVPA volume (compared to graphs D-F). This is what is stated in the figure title but in lines 101-102 it is stated that the third model is additionally adjusted for sleep duration, sleep phase, as well as total MVPA volume (and sleep duration and sleep phase are listed in model 2 for graphs D-F).

Response 9: **Fig. 1** incorporates the fully adjusted associations of total MVPA volume [(a)-(c)], MVPA within time windows (Continuous) [(d)-(f)], and fractions of MVPA within time windows [(g)-(i)] with mortality risk. This figure should facilitate understanding and comparisons of the potential health effects of MVPA (total and during different time of day) and MVPA timing. Therefore, for different associations in **Fig. 1**, the fully adjusted models need to include different covariates.

Graphs **(d)-(f)** show the associations between MVPA within three time windows and mortality outcomes after controlling for age, sex, ethnicity, Townsend index of deprivation, recruitment center, education level, season of accelerometer wear, smoking status, alcohol intake, healthy diet score, sleep duration (< 7 hours, 7-8 hours, > 8hours), sleep midpoint, and MVPA volume during other two time windows (~ modified model 3, **not model 2**). Graphs **(g)-(i)** show the associations between the fractions of MVPA within three time windows with mortality risk after adjusting for age, sex, ethnicity, Townsend index of deprivation, recruitment center, education level, season of accelerometer wear, healthy diet score, smoking status, alcohol intake, sleep duration (< 7 hours, 7-8 hours, > 8hours), sleep midpoint, and total MVPA volume (model 3). Therefore, the relationship between graphs **(g)-(i)** and **(d)-(f)** is not like model 3 vs. model 2. We hope that the reviewer is satisfied with our clarifications.

Comment/Question 10: -Line 225. I think you mean “test” instead of “testify”.

Response 10: According to the reviewer’s suggestion, we have changed “testify these hypotheses” to “test these hypotheses”.

Comment/Question 11: -Line 245. Regarding potential limitations, how robust is your estimate of MVPA timing? Is 1-week behavior reflective of longer-term behavior? If not, how might this influence the interpretation of these results based on only 7-days of data?

Response 11: We thank the reviewer for bringing this concern to our attention. We provided a more in-depth discussion in the limitation part as follows:

“Third, although 7-day monitoring periods have been routinely used in PA monitor studies because they provide the opportunity to sample PA on both week and weekend days and achieve intra-class correlations of greater than 80% in most populations³⁵, it is still unclear whether a 7-day accelerometer measurement is representative of longer-term behaviors³⁶, especially for PA timing. In addition, although our findings were robust to daylight saving time and adjustment of season (even month) of accelerometer wear, these sensitivity analyses may not be sufficient to account for potential misclassification or seasonal variation of PA timing. These situations highlight the importance of repeated and long-term administration of accelerometers in future studies.” [Discussion (page 8; line 186-193)]

Reviewers' Comments:

Reviewer #1:

Remarks to the Author:

I thank the authors for a very comprehensive response to my initial concerns and questions to their manuscript. The results are noteworthy and will be of significance to the field of public health and physical activity.

Reviewer #2:

Remarks to the Author:

Thank you for your thorough response to the suggested edits/comments from this and the other reviewers. The manuscript is improved and provides sufficient detail in methods, data interpretation, and discussion of the significant findings/limitations. The manuscript could be strengthened by providing a more succinct summary/conclusion at the close of the discussion. Perhaps with a bit more specificity regarding the types of 'experimental' studies needed to confirm hypotheses regarding the timing of daily physical activity participation and subsequent health outcomes (NCDs, functional mental/physical health).

Reviewer #3:

Remarks to the Author:

The authors have made substantial beneficial modifications to their manuscript in response to all reviewers comments. In particular, they have satisfactorily addressed my specific comments. I commend them on their excellent work.

**** Response to Reviewers' comments ****

Reviewer #1 (Remarks to the Author):

I thank the authors for a very comprehensive response to my initial concerns and questions to their manuscript. The results are noteworthy and will be of significance to the field of public health and physical activity.

Response to reviewer #1: We thank the reviewer for his/her professional assessment and positive comments.

Reviewer #2 (Remarks to the Author):

Thank you for your thorough response to the suggested edits/comments from this and the other reviewers. The manuscript is improved and provides sufficient detail in methods, data interpretation, and discussion of the significant findings/limitations. The manuscript could be strengthened by providing a more succinct summary/conclusion at the close of the discussion. Perhaps with a bit more specificity regarding the types of 'experimental' studies needed to confirm hypotheses regarding the timing of daily physical activity participation and subsequent health outcomes (NCDs, functional mental/physical health).

Response to reviewer #2: We appreciate the reviewer's positive comments. According to the reviewer's suggestions, the end of the discussion section has been modified as follows:

“In summary, our results show that MVPA timing may have the potential to maximize the health benefits of daily PA. Future experimental studies in humans are needed to confirm the impact of the timing of various types of PA on subsequent health outcomes.” [Discussion (page 9; lines 230-232)]

Reviewer #3 (Remarks to the Author):

The authors have made substantial beneficial modifications to their manuscript in response to all reviewers comments. In particular, they have satisfactorily addressed my specific comments. I commend them on their excellent work.

Response to reviewer #3: We appreciate the reviewer's positive comments. The comments and suggestions from the reviewers have substantially improved our manuscript.